# Child volunteers in a women's paramilitary organization in World War II have accelerated reproductive schedules

Robert Lynch 1✉, Virpi Lummaa1, Michael Briga 1, Simon N. Chapman 1 & John Loehr 2

Understanding how conditions experienced during development affect reproductive timing is of considerable cross-disciplinary interest. Life-history theory predicts that organisms will accelerate reproduction when future survival is unsure. In humans, this can be triggered by early exposure to mortality. Previous studies, however, have been inconclusive due to several confounds that are also likely to affect reproduction. Here we take advantage of a natural experiment in which a population is temporarily divided by war to analyze how exposure to mortality affects reproduction. Using records of Finnish women in World War II, we find that young girls serving in a paramilitary organization wait less time to reproduce, have shorter inter-birth intervals, and have more children than their non-serving peers or sisters. These results support the hypothesis that exposure to elevated mortality rates during development can result in accelerated reproductive schedules and adds to our understanding of how participation in warfare affects women.

---

[1] University of Turku, Turku, Finland. [2] University of Helsinki, Helsinki, Finland. ✉email: robertflynch@gmail.com

Understanding the variation in women's reproductive scheduling among individuals, populations, age groups, and environments is of considerable interest to researchers across disciplines. Life-history theory provides a general framework for understanding how organisms allocate time and energy into reproduction in different environments to maximize their fitness[1]. Trade-offs between investing in growth, somatic maintenance, and reproduction, for example, can result in different optimal strategies with respect to reproductive timing or investment in offspring[2]. Although environmental stress and adversity are often seen as being detrimental to fitness under all conditions[3], models generated out of life-history theory predict that individuals will adjust their reproductive strategies adaptively in response to conditions they experience during development[4]. Reproductive strategies ultimately depend on the fitness returns for producing and investing in children in the face of an uncertain future, and there is considerable theoretical[5], empirical[6], and experimental[7] support for the hypothesis that both harsh and unpredictable environments result in greater early age reproductive effort and reduced parental investment, when the chances of one's own future survival and reproduction are uncertain. Therefore, life-history theory sees conditions in early-life as signals that can trigger adaptive responses which enable organisms to maximize their fitness in changing environments[8].

Research on the impact of developmental conditions on human reproductive timing has often focused on stress. For example, children who experience higher levels of familial stress have been shown to have earlier first births[8] and stressful childhoods have been positively associated with both earlier age of menarche[9] and pregnancy[10]. Reduced parental investment, particularly father absence, has also been shown to affect reproductive timing. A meta-analysis revealed a strong association between father absence and earlier menarche, and the experimental priming of father disengagement has been shown to predict increased sexual risk taking behavior[11]. Results of a study using data from the British National Child Development Study showed a positive relationship between low father involvement and earlier age at first birth[12]. However, a recent meta-analysis suggests these results may be restricted to certain populations[13]. Similarly, childhood trauma has also been hypothesized to affect reproductive timing. One study showed that girls between the ages of 4 and 11 who were separated from their parents during the evacuation of Helsinki during World War II had earlier menarche and more children by late adulthood than those who remained with their families[14]. However, another study comparing same-sex siblings from the same population did not find any relationship suggesting that the first result was largely due to selection bias: evacuees came from poorer and larger families which affected their reproductive behavior[15]. This illustrates a common difficulty in inferring causality in studies seeking to determine the impact of complex and frequently interrelated developmental conditions on human reproductive behavior.

The most theoretically plausible environmental cues hypothesized to affect reproductive timing are those that involve sensitivity to local mortality rates[2,16]. In low mortality environments, parents may be better off by investing in their own growth and survival or investing more in current offspring[17]. Both between and within mammalian species, mortality rate is the best predictor of life-history traits, and higher juvenile mortality in particular has been positively correlated with earlier maturation, faster development, smaller offspring, and higher lifetime reproduction[18]. Evidence for later first birth in low mortality environments[5] and earlier reproduction in high mortality environments[19] suggests that humans also respond to mortality cues. Responses can be triggered by cataclysmic events; in Iceland, for example, historical records show that reproductive rates

increased in the aftermath of volcanic eruptions[20,21]. However, in one indigenous community birth rates fell after the fall of the Soviet Union[22] suggesting that individuals may at times employ a more conservative wait-and-see strategy while absorbing information about a novel environment. It remains unknown which specific cues individuals use to estimate local mortality, but some have argued that, at the psychological level, assessing this risk affects discounting rates—the relative value individuals place on present versus future rewards[8].

Multiple cues can also interact to influence reproductive decisions[23], which makes isolating a particular variable from other potential causes using correlative data difficult, and social class is a particular concern. One study showed that individuals who claimed to have been raised in high mortality rates, low resources environments reported preferences for reproducing earlier and having larger families, while individuals who claimed to have been raised in low mortality, high resource environments had the opposite preferences[24]. Another study, however, found that adjusting for social class and education eliminates these associations[25]. Overall, a review of research on the relationship between early life adversity and reproductive timing in humans has shown qualified support for the hypothesis that stress early in development results in accelerated reproductive schedules, but was inconclusive with regard to its effect on total fertility[10]. Although stronger conclusions could be drawn using new approaches which are better able to control for potential confounds and which examine both reproductive timing and total reproduction, these data are rare in humans.

Regardless of which cues humans respond to when adjusting life-history strategies, many suggest that signals received during childhood are expected to have the greatest impact[9,26]. This is because childhood provides the necessary time to effectively plan and adjust reproductive strategies in response to both one's own biological condition[27,28] and the social environment[8,29]. Evidence showing that individuals who reported being raised in harsh environments expressed preferences for earlier reproduction when exposed to cues indicating mortality rates were increasing[7] supports this idea that reproductive decisions are moderated by environments experienced during childhood. It is important to note, however, that other experimental studies have failed to provide support for this hypothesis, and a review article suggested that overall causation has yet to be established[30]. Wilson and Daly[19] also demonstrated that factors affecting reproductive timing may be age-specific: although young women living in Chicago neighborhoods with the lowest life expectancy's reproduced earlier than women in neighborhoods with high life expectancy, this effect disappeared by age 30.

In this study, we take advantage of a natural experiment in which a population of evacuees is separated by World War II into two groups, each differentially exposed to mortality cues and stress, and then reintegrated back into the same population when the war ends to analyze how participating in war affects reproduction. Specifically, we use an unusually well-documented dataset of 37,613 Finnish women who were evacuated from Karelia during World War II to compare the age-specific reproductive timing and success of volunteers for a woman's paramilitary group in Finland called Lotta Svärd—an organization tasked with supporting troops as nurses, air raid spotters, mess personnel and used in other auxiliary capacities[31]—with their peers and sisters who did not volunteer. Although previous research has shown that exposure to stress can affect reproductive schedules, much of it has failed to control for key variables, such as father absence, socioeconomic status, effects of family, and shared, but frequently unknown demographic variables within comparison groups. Here, we use a large demographically diverse database of volunteers from the same population, neighborhoods,

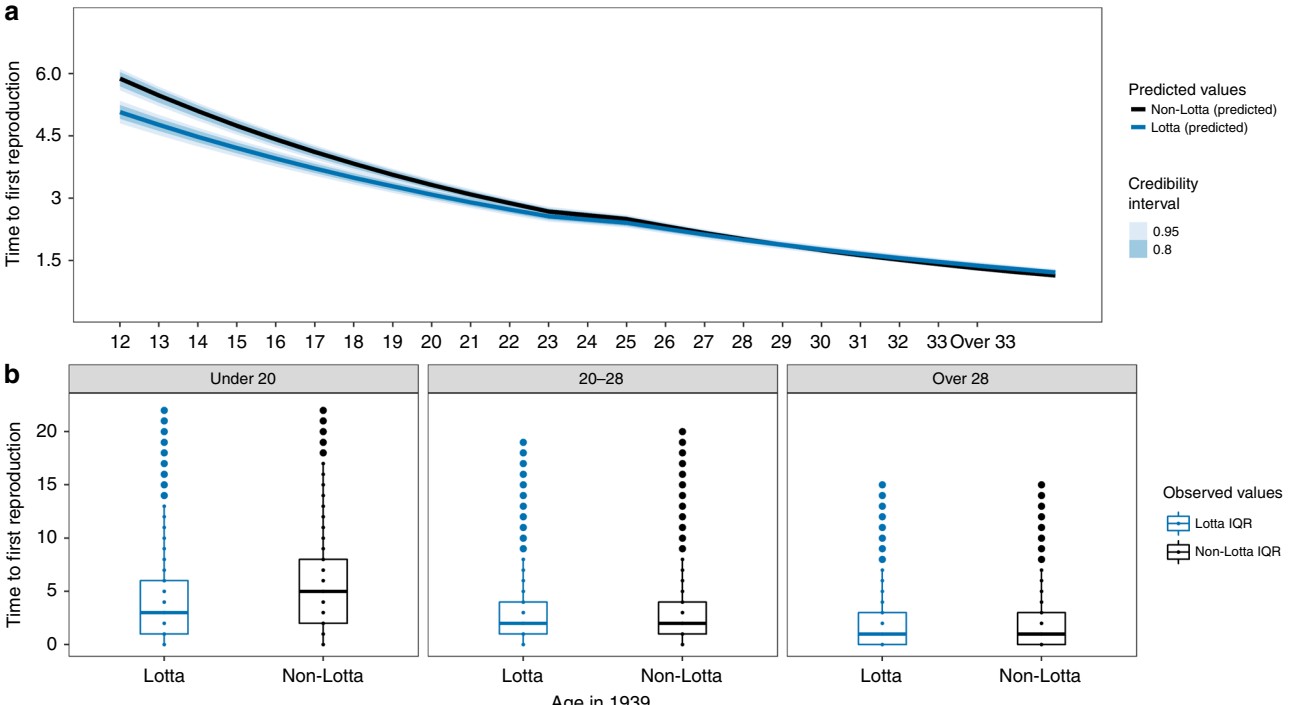

**Fig. 1 Younger volunteers of the same age as women who did not volunteer waited less time to give birth after the war ended in 1945. a** Model-generated posterior distribution predictions (dark lines), credibility intervals (shaded). **b** The observed data (median—solid line, box—interquartile range (25% and 75%), whiskers—5% and 95% intervals) for years to reproduction (y-axis) for age groups "Under 20", "20–28", and "Over 28" when the war began in 1939. See Supplementary Materials: Fig. 1a and Table 1 (top panel, right side) for posterior distributions for all covariates and Supplementary Fig. 3a for posterior predictive check for this model. Differences between the model-generated predictions in panel **a** and the observed data (panel **b**) primarily result from the impact of covariates entered into the model.

backgrounds, ethnic groups, and families as women who did not volunteer, and which also includes key variables that previous research on this population has identified as having important effects on fertility[32]. These data are particularly useful for examining the effects of exposure to mortality, stress, and trauma during war on the reproductive schedules of young women for four main reasons. First, most studies exploring the effects of war on reproduction have focused on men[33,34]. Second, these data are unique in their detailed and comprehensive recording of the life-histories, marriages, occupations, and war service records of an entire population of war evacuees. Third, because we have data on the service records and future occupations of all of these women, we are able to disentangle the effect of social class from the effects of exposure to mortality and war-induced stress on reproductive schedules. Finally, we were able to link some of these data to genealogical and interview records to analyze a subset of sisters who came from the same families. Specifically, we predict that after the war ends, younger volunteers who were exposed to higher mortality (**P1**) will wait less time to have a child, (**P2**) have shorter interbirth intervals (IBI), and (**P3**) higher lifetime reproductive success than their age-matched peers and sisters who are not exposed to these conditions.

## Results

**Younger volunteers have faster reproductive schedules**. We used a Bayesian generalized linear mixed-effects (GLMM) regression using the rethinking package[35] in R Studio 3.3.3 to analyze the reproductive timing and lifetime reproductive success of Lotta Svärd volunteers. Overall, results showed that girls who were exposed to higher mortality when they were younger had accelerated reproductive schedules and higher

overall reproductive output after the war ended. This was true both for models run on the full sample of evacuees [$N = 37,613$ and $N = 31,613$ for all women and only women who reproduced, respectively] and for a subset of individuals who we were able to link to a genealogical database and who had at least one full sister [$N = 2671$ and $N = 2272$ for all women and only women who reproduced, respectively] (see Methods). The model predictions, results and raw data for age-specific reproductive schedules and output are shown in Figs. 1–4 and Supplementary Materials: Figs. 1a, b, 2a, b, and Supplementary Tables 1 and 2. Error bars and raw data for all variables used in this study can be accessed with this interactive app: https://www.helsinki.fi/en/projects/learning-from-our-past/data#section-60700.

We found support for the prediction that women exposed to higher mortality waited less time to reproduce after the war ended given their higher early life exposure to mortality rates, compared to peers from the same background and neighborhoods not exposed to these rates (**P1**). Using the full sample of evacuees [$N = 31,613$] the model predicts that a volunteer who was 15 years old when the war broke out waited an average of 5.44 (95% PI: 5.21–5.67) years until they had their first child after the war ended. This is two-thirds of a year less than the prediction for a 15-year-old who did not volunteer (6.35 years, 95% PI: 6.13–6.57). This pattern holds if predictions are limited to women who all had their first child only after the war ended (5.66 for Lottas vs. 6.61 for non-Lottas). There was, however, no detectable effect of volunteering on the time to first reproduction for older women. Women who were 30 years old when the war began were predicted to wait an average of 2.01 (95% PI: 1.92–2.10) and 2.02 (95% PI: 1.95–2.10) years for Lottas and non-Lottas respectively (see Fig. 1a, b and Supplementary Materials: Table 1—top panel, right side—and Fig. 1a). Using a subset of

evacuees whose parents were known and who had at least one sister [$N = 2272$], the model predicts that a volunteer who was 15 years old when the war broke out would have waited an average of 4.65 (95% PI: 4.25–5.08) years until they had their first child after the war ended. This is two-thirds of a year less than the prediction for a 15-year-old who did not volunteer (5.32 years, 95% PI: 4.90–5.75). The opposite pattern is seen for women who were 30 years old when the war began, who were predicted to wait an average of 0.37 years longer—4.23 (95% PI: 3.97–4.51) and 3.86 (95% PI: 3.62–4.12) years for Lottas and non-Lottas, respectively (see Fig. 3a, b and Supplementary Materials: Table 2 —top panel, right side—and Fig. 2a).

Next, we investigated the prediction that post-war birth intervals will be shorter for Lottas who volunteered when they were younger (**P2**). This prediction was also supported: Using the full sample of evacuees [$N = 31,613$] the model predicts that a volunteer who was 15 years old when the war began will have a mean postwar birth interval of 4.13 (95% PI: 4.07–4.20) years, which is 2 months shorter than for girls who did not volunteer with a mean birth interval of 4.34 (95% PI: 4.32–4.36). The pattern is the same when we limit our predictions to 15-year-old women who had their first child after the war ended (4.23 for Lottas and 4.44 for non-Lottas). The difference in mean birth intervals between volunteers and women who did not volunteer again vanishes for older women. Women who were already 30 years old when the war began are predicted to have a mean birth interval of 2.14 (95% PI: 2.08–2.21) years for Lottas and 2.18 (95% PI: 2.15–2.21) years for non-Lottas after the war (see Supplementary Materials Table 1—middle panel, right side). This prediction, however, received only slight support from the subset of evacuees whose parents were known and who had at least one sister [$N = 2272$]. Here, the model predicts that a volunteer who was 15 years old when the war broke out would have had a mean post-war birth interval of 5.58 (95% PI: 4.39–6.96) years, which is nearly identical to the predicted birth interval of 15-year-old girls who did not volunteer 5.56 (95% PI: 4.46–6.65) years. Older volunteers (i.e., women who were 30 years old when the war began), however, were predicted to have somewhat longer postwar birth intervals—4.07 (95% PI: 3.60–4.55) and 3.85 (95% PI: 3.43–4.30) years for volunteers and non-volunteers, respectively (see Supplementary Materials: Table 2—middle panel, right side) which is consistent with prediction (**P2**), but does not offer strong support of it. It is important to recognize, however, that our models of time to first reproduction and average IBI only include women who reproduced. Therefore we conducted sensitivity analysis[36] to determine the impact of excluding non-reproductive women from these models and found that overall results were very similar (see Supplementary Results: Sensitivity analysis, Supplementary Fig. 5 and Supplementary Tables 3 and 4).

Finally, we tested the prediction that accelerated reproductive schedules among younger volunteers will result in higher total post-war reproductive success (**P3**). Using the full sample of evacuees [$N = 37,613$] we found some support for this prediction: using all women who were between the ages of 12 and 39 in 1939 (i.e., aged 43–70 when the interviews took place in 1970), including those who never reproduced at all, the model predicts higher postwar reproductive success for volunteers who were less than 22 years old when the war started. For example, 15 year olds who volunteered are predicted to have an average of 1.93 (95% PI: 1.83–2.02) children after the war, while those who did not are predicted to have 1.84 (95% PI: 1.75–1.93) children on average. The pattern is the same when we limit our predictions to women who had their first child after the war ended (2.85, 95% PI: 2.79–2.91 for Lottas and 2.72, 95% PI: 2.66–2.78 for non-Lottas). This does not seem to be true, however, for women who were

already 30 years old when the war began (1.08, 95% PI: 1.02–1.14 for Lottas vs. 1.25, 95% PI: 1.19–1.31 for non-Lottas) (see Fig. 2a, b and Supplementary Materials Table 1: bottom panel, right side). We also tested this prediction using a subset of evacuees whose parents were known and who had at least one sister [$N = 2671$]. Although results were in the predicted direction, they do not offer strong support for the hypothesis. Here, the model predicts that a volunteer who was 15 years old when the war broke out would have 1.19 (95% PI: 0.70–1.85) children after the war ended, which is only slightly more than the 1.11 (95% PI: 0.66–1.68) children 15-year-old girls who did not volunteer are predicted to have. A stronger, opposite pattern, however, is seen for older volunteers, whereby volunteers were predicted to have fewer children after the war than non-volunteers—0.94 (95% PI: 0.56–1.44) and 1.18 (95% PI: 0.71–1.75) children after the war for 30-year-old Lottas and non-Lottas, respectively (see Fig. 4a, b and Supplementary Materials: Table 2—bottom panel, right side—and Fig. 2b).

Several additional models were also run in an attempt to further parse the effects of volunteering on reproductive outcomes. First, because these women were not randomly assigned (i.e, they volunteered), any number of unmeasured differences between Lottas and non-Lottas that could affect reproductive timing are possible. We sought to control for this type of selection bias by entering key covariates into the models, analyzing a subset of sisters (see "Methods: Statistical Analysis") to control for family effects, and comparing Lottas and non-Lottas across a variety of potentially relevant traits (Supplementary Materials: Table 6). Most importantly, however, we were able to analyze the reproduction of many of these same women before the war to determine more directly the impact that the war had on their fertility. A comparison of these models analyzing the age based reproductive outcomes of women before vs. after the war provides additional support for the hypothesis that experiences during the war accelerated the reproductive schedules of young volunteers and strongly suggests that these results are not produced by selection bias (see Supplementary Results: Selection bias, Supplementary Tables 1 and 7).

Additional models were also run to distinguish between other plausible hypotheses. For example, because these results could have been driven by the fact that volunteers were likely to have had more contact with soldiers, we also analyzed the effect of marriage on accelerated reproductive schedules and found that this possibility is unlikely because these results also hold for volunteers who were married before the war (see Supplementary Results: Effect of exposure to male soldiers and Supplementary Table 7). Finally, in an attempt to further determine how particular wartime activities might differentially affect reproduction we categorized volunteers as being more or less exposed to combat for a small subset of women who reported their division within Lotta Svärd. Although the trend was in the hypothesized direction (i.e., young volunteers who were presumably more exposed to combat had marginally faster reproductive schedules) results were not significant (see Supplementary Results: Lotta type and Supplementary Table 5).

## Discussion
Theory and data suggest that humans use mortality cues observed during development to adaptively regulate reproductive timing and effort[5,17,19]. Much of this research has used broad demographic data on neighborhoods in contemporary societies[19], or experimental primes[7] to investigate how differential exposure to local mortality rates might cause individuals to adjust their reproductive schedules. To our knowledge, however, no study has used a natural "experiment" in which a single population is temporarily split into two groups, each differentially exposed to

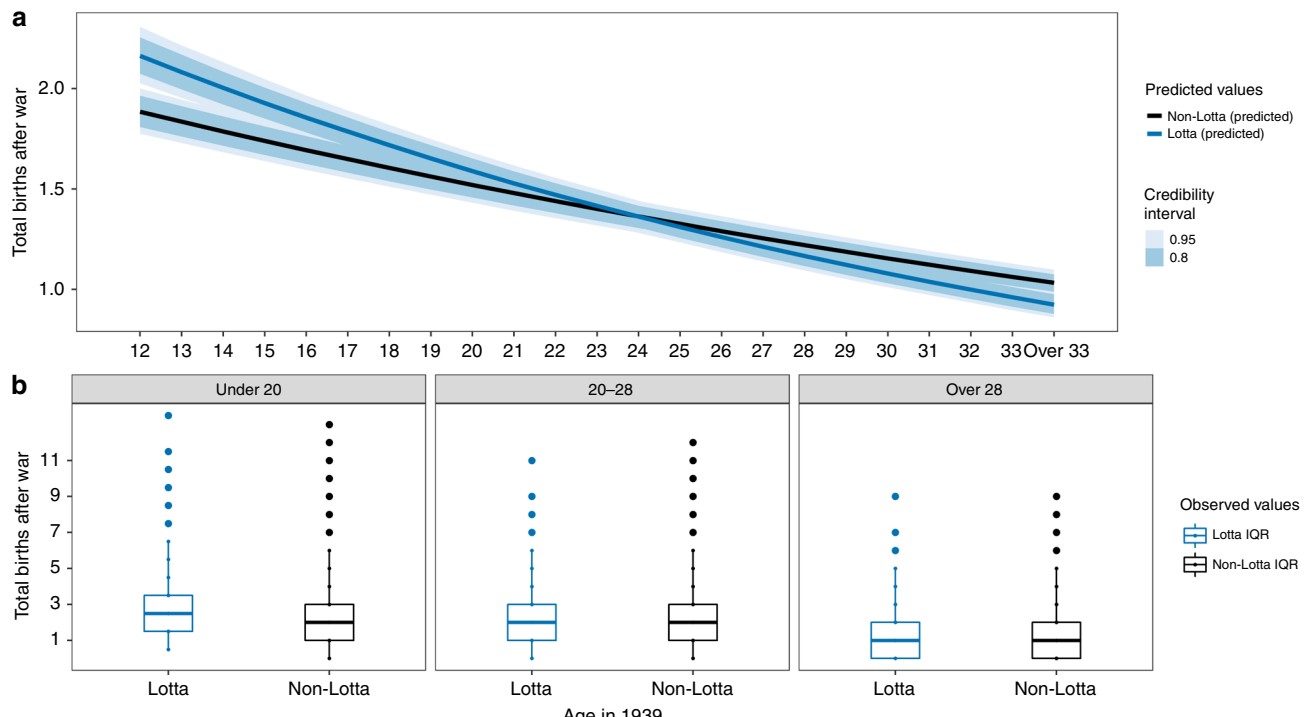

**Fig. 2 Younger volunteers of the same age as women who did not volunteer were predicted by the model to have slightly more children after the war ended in 1945. a** Model-generated posterior distribution predictions (dark lines), credibility intervals (shaded). **b** The observed data (median—solid line, box—interquartile range (25% and 75%), whiskers—5% and 95% intervals) for total reproduction after the war (y-axis) for age groups "Under 20", "20–28", and "Over 28" when the war began in 1939. See Supplementary Materials: Fig S1b and Table S1 (bottom panel, right side) for posterior distributions for all covariates and Fig. S3b for Posterior predictive check for this model. Differences between the model-generated predictions in panel **a** and the observed data (panel **b**) primarily result from the impact of covariates entered into the model.

mortality and stress conditions, to analyze how young girls participating in war affects reproductive rates. Our results provide strong support for the hypothesis that women strategically adjust reproduction in response to environmental conditions, and suggest that female reproduction is sensitive to local mortality rates during development. These findings are of general relevance for multidisciplinary efforts to understand how inequality in health care, crime rates, war, and broad differences in exposure to mortality rates link to changing patterns of teen pregnancies, postponement of first reproduction, and declining fertility rates across the world.

We found that young women who volunteered for a paramilitary organization during World War II had accelerated reproductive schedules and higher overall lifetime reproductive success. Specifically, Lotta Svärd volunteers who were younger during the war waited less time to have their first child, had shorter IBI and had more children after the war ended in 1945. Analyses of the full population-based sample of evacuees, and a subset of women who had at least one sister and controlling for family effects (e.g., shared parents, households and genetics among siblings), yielded similar results. This study made use of a dataset containing the life-history records of a rare population—young girls participating in war. Although previous research on the effect of traumatic or stressful childhood experiences on future reproduction has shown similar results, much of it has failed to control for key variables, including father absence, socioeconomic status, and shared, but unknown, demographic variables within comparison groups. Here, we took advantage of a quasi experiment in which the sudden onset of war divided a population into two groups. Each group was differentially exposed to stress, trauma, and mortality, and then reintegrated back into the same population when the war ended. Although

these Lotta Svärd volunteers were a demographically diverse group who came from the same neighborhoods, backgrounds, ethnic group and population as the women who did not volunteer, we were still able to control for key factors such as education and working in agriculture, which previous research on this population has identified as having important effects on fertility[32].

Investing in current vs. future reproduction, also known as slow vs. fast life-histories, is a fundamental trade-off all organisms face, and optimal investment strategies are expected to depend on the environmental conditions individuals encounter during development[9]. In humans, some individuals adopt slower strategies, characterized by later reproductive development, delayed sexuality, stronger preferences for monogamy, high parental investment, and an orientation toward future outcomes, while others display faster strategies characterized by the opposite[37]. Although our results may be best understood within the framework of life-history theory (e.g., stressful conditions or exposure to high local mortality rates lead to accelerated reproduction), they may also be considered more broadly under time discounting. Discounting is the relative value individuals place on present vs. future rewards[38], and it is frequently tied to extrinsic mortality rates[39].

The mechanisms by which harsh environmental conditions are likely to hasten reproduction, or, on a psychological level, increase discounting rates, are currently debated, but exposure to stress[40] and elevated local mortality rates[1,19,41] are both widely seen as strong predictors of earlier reproduction even if the timing of these effects is still debated[42]. The evolutionary logic here is clear: if organisms perceive a higher probability of dying, then they are better off allocating resources toward reproductive output and less toward parental investment, in order to maximize

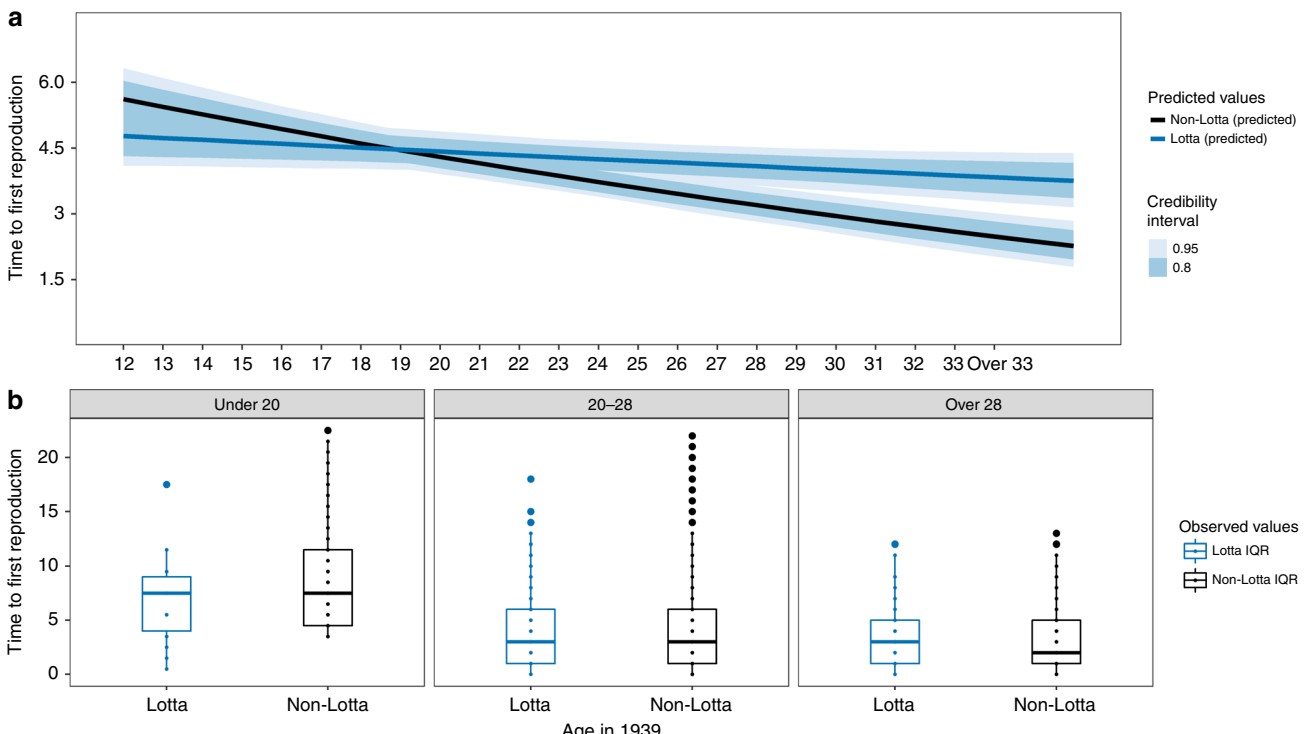

**Fig. 3 In the sisters only analysis (see Methods: Statistical Analysis) the effect of volunteering on time to reproduction after the war was also age specific such that younger volunteers waited less (and older volunteers more) time to give birth than their sisters who did not volunteer. a** Model-generated posterior distribution predictions (dark lines), credibility intervals (shaded). **b** The observed data (median—solid line, box—interquartile range (25% and 75%), whiskers—5% and 95% intervals) for years to reproduction (y-axis) for age groups "Under 20", "20–28", and "Over 28" when the war began in 1939. See Supplementary Materials: Fig. 2a and Table 2 (top panel, right side) for posterior distributions for all covariates and Supplementary Fig. 4a for Posterior predictive check for this model. Differences between the model-generated predictions in panel **a** and the observed data (panel **b**) primarily result from the impact of covariates entered into the model.

the chances that at least some offspring will survive[1,7,12]. But how exactly do childhood adversity and exposure to mortality affect reproductive strategies? On a behavioral level, these conditions may increase tolerance for risk. Risk taking, for example, has been positively associated with childhood stress, which has in turn been linked to expectations of a shorter lifespan and an unpredictable future[43]. At the same time, the experimental priming of higher local mortality rates has been shown to increase the desire for children and babies[44]. Furthermore, changing perceptions of mortality rates not only alter one's expectations of survival, but can also affect ones future decisions and life course[45].

Although we were unable to distinguish between the impacts of stress and exposure to higher mortality because Lottas were likely to have experienced higher levels of both, we are able to dismiss several other commonly cited mechanisms as highly unlikely. First, father absence is not expected to have had an important impact on these results, because volunteers from the same families who had the same fathers—sisters—also had accelerated reproductive schedules. Second, socioeconomic status is unlikely to have had a significant impact because we controlled for agricultural occupation, education, and parents. Finally, countless additional factors that are likely to impact neighborhoods or populations in different ways are less likely to affect our results than in previous demographic studies. This is because Lottas were drawn from the same neighborhoods, families and ethnic group as non-Lottas and then exposed to the war at different levels for a limited period of time, before returning to similar conditions after 1945, when our analyses begin. Controlling for these variables is particularly important because there are likely to be crucial interactions among both known (e.g., father absence and

poverty) and unknown environmental cues. For example, although women in populations with higher infant mortality have earlier first births[19,46], these populations are frequently associated with higher poverty. Because lower socioeconomic status and high mortality are both seen to predict higher fertility, earlier age of first reproduction, shorter IBI, and lower parental investment[47,48], it can be extremely difficult to tease apart their effects.

Still controlling for these key variables may not be enough because there may be unknown characteristics of certain families that both increase the likelihood of volunteering and result in accelerated reproductive rates. This type of biased sample can lead to spurious conclusions due to shared parents, environments and genetics among siblings (see ref. [15]) on child evacuees from Helsinki in World War 2. Therefore, it is important to try to control for these effects. For example, although within families women who volunteered had both accelerated reproductive schedules and higher overall reproduction, the effect sizes were not as high as they were with the full sample which suggests that we should be cautious in overinterpreting these results. The analysis of sisters does have its disadvantages, however. For example, the lower effect size found in the sister analysis may be due in part to a lower sample size and an overall reduction in statistical power. Furthermore, limiting our analysis to only include families with at least two daughters can introduce bias by excluding singletons, or those with only brothers. The side by side analysis of the traits of women who volunteered versus those who did not for the full sample of evacuees and for the subset of sisters also reveals some of the key advantages (e.g., sisters are much closer than non sisters on most of the traits) and disadvantages

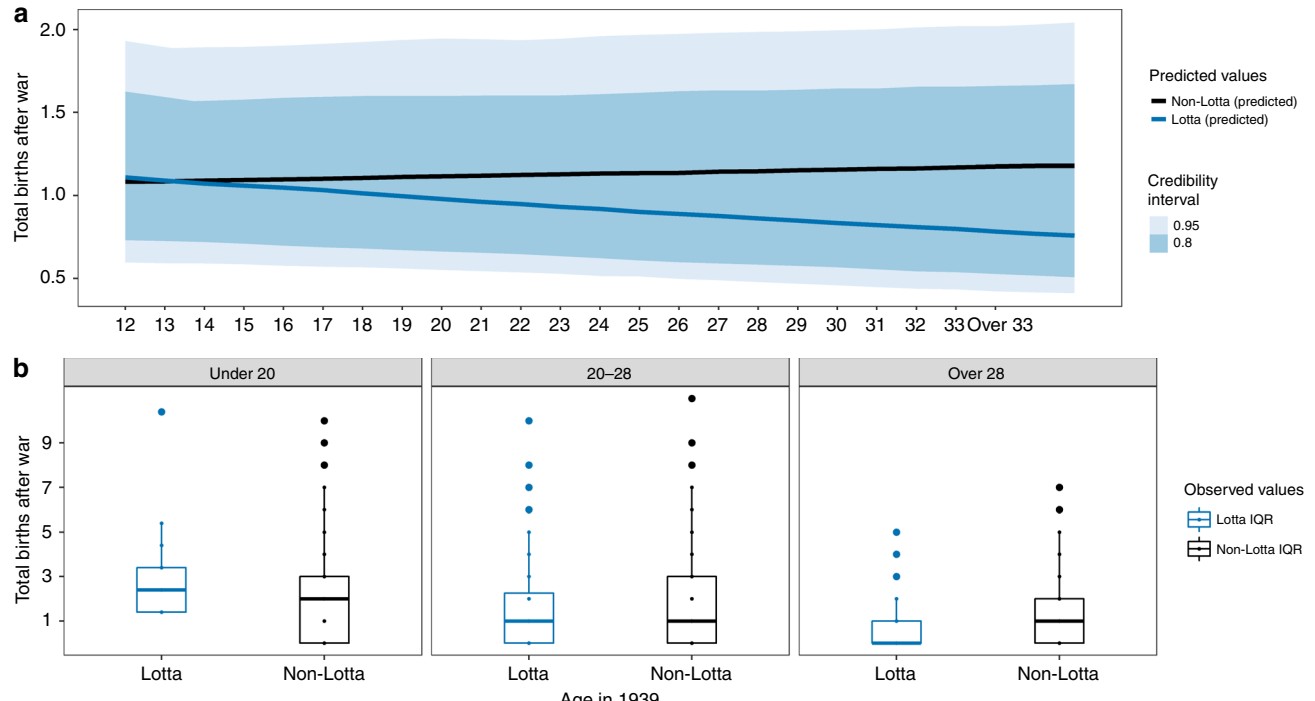

**Fig. 4 In the sisters only analysis (see Methods: Statistical Analysis) the effect of volunteering on total reproduction after the war was also age specific such that younger volunteers had slightly more (and older volunteers slightly fewer) children than their sisters who did not volunteer. a** Model-generated posterior distribution predictions (dark lines), credibility intervals (shaded). **b** The observed data (median—solid line, box—interquartile range (25% and 75%), whiskers—5% and 95% intervals) for total reproduction after the war (y-axis) for age groups "Under 20", "20–28", and "Over 28" when the war began in 1939. See Supplementary Materials: Fig. 2b and Table 2 (bottom panel, right side) for posterior distributions for all covariates and Fig. 4b for posterior predictive check for this model). The main effect of the model generated predictions here is that younger volunteers have similar reproductive outcomes as non-volunteers while older volunteers are predicted to have fewer children. Differences between the model-generated predictions in panel **a** and the observed data (panel **b**) primarily result from the impact of covariates entered into the model.

(e.g., sample size and higher standard errors) of limiting our analysis to only sisters. There are also some basic limitations of using these type of historical data. For example, we do not have genetic data or personality traits or a nearly endless variety of other potential differences between women who volunteer and those who do not that may have an effect on age sensitive reproductive timing. Although analyzing sisters and controlling for many potential family effects is a big step in the right direction, there are still any number of possible differences of which we are unable to take into account without conducting a controlled experiment. Still it is important to recognize that if this effect were primarily social (e.g., Lottas are more gregarious), it is unclear why it would only be manifested in volunteers who were young and only before the war began. Finally, although modeling the reproductive schedules of women before the war and comparing these results to many of these same women after the war marks a substantial improvement over the type of selection bias issues that have plagued previous quasi-experimental studies on natural populations, they are still imperfect. This is because, even though there is substantial overlap between the samples they are not identical for the models assessing time to first reproduction and IBI (see Methods: "Statistical Analysis"). Nevertheless, the fact that all of the women are included in the postwar sample and a dummy variable marking whether they had given birth prior to the war was used in these models lends more credibility to the hypothesis that exposure to mortality had an effect on these results. Furthermore, it is just as important to recognize that for overall reproduction the exact same sample of women was used both before and after the war (S1 and S2—bottom panel), making more precise comparisons possible.

It is also unlikely that increased exposure to mortality is the only factor causing young volunteers to have faster reproductive schedules, and young volunteers being more exposed to soldiers is one plausible alternative explanation. Indeed, previous research on this same population has shown that women who had more brothers and whose husbands served in the military were more likely to volunteer[49]. Models which included whether or not these women were married when the war began confirmed that the Lotta by age interaction was still a significant predictor of faster reproductive schedules. But they also indicate that the combination of being married and being a Lotta results in faster reproductive schedules after the war (Supplementary Materials: Table 7). This suggests that volunteers who were already married before the war began also had faster reproductive schedules. Because the reproductive outcomes of married women are less likely to be accelerated by more interactions with men, these results are unlikely to be entirely driven by greater exposure to soldiers. In addition, the primary effect of accelerated reproductive schedules is for Lottas who were between the ages of 12 and 20 (see Figs. 1–4), while the mean age at marriage in our sample is 26.2 years old (6–14 years after the war began for these women). Therefore, it is unlikely that meeting men with whom they were likely to have had more contact can fully explain these results. Our analysis of different types of volunteers based on their presumed exposure to combat did reveal a trend in the hypothesized direction (i.e., younger women in Lotta units that were more exposed to mortality had faster reproductive schedules). However, the models failed to detect a significant interaction between Lotta exposure and age for any of the three dependent variables assessing reproductive rate after the war. Although our inability

to detect significant differences between these Lotta groups can be written off as the result of severe sample size restrictions (less than 10% of Lottas reported their units), our assumptions about which Lottas were exposed to higher mortality and/or a considerable amount of noise between categories, it does provide an additional reason to be cautious in overinterpreting these results.

The impact of environmental conditions experienced during development on adult growth, development, reproduction, and behavior is central to many disciplines, including evolutionary biology, ecology, ethology, developmental health, social psychology, and sociology[50–55]. In this study we are able to rule out many cues that have previously been hypothesized to signal harsh environments or reduced parental investment while controlling for a number of confounding variables that have conspired to make drawing strong conclusions difficult. Overall these results add to the growing literature on the impact of early life conditions on subsequent reproductive strategies and suggest that women adaptively respond to conditions they experience in early life. Because this study was conducted on war evacuees, these results may also be of particular relevance to researchers interested in the impact of violent conflict on refugees and understanding the effects of war on life-history traits such as reproduction may be an important consideration when attempting to mitigate the harmful effects of these conflicts.

## Methods

**Historical background**. All of the individuals in this study were evacuated from Finnish Karelia during the Second World War. The Soviet Union invaded Finland in 1939, starting the Winter War. As part of the Moscow Peace Treaty of 1940, Finland ceded territory, including Karelia, to the Soviet Union, and evacuated the entire population to the rest of Finland. Many evacuees moved back[49] during the Continuation War (1941–1944), which saw Finland briefly reclaim this lost territory. However, the Soviet Union once again conquered Finnish Karelia in 1944, and the territory permanently moved into Soviet possession with the signing of Moscow Armistice agreement. Again, the population that had returned was evacuated and settled in western Finland.

**Lotta Svärd organization**. Founded in 1920, the Lotta Svärd organization was a volunteer paramilitary organization for women which provided much needed military support to the Finnish armed forces. Members operated at the front lines as well as on the home-front in various duties included nursing, food service, anti-aircraft spotting, fundraising, and messenger activities. The youth corps was created in 1931 for children aged 8–16, with 14–16 year olds taking on duties with greater responsibility. In total, there were 221,613 volunteers in the adult and youth corps by the end of the war[56].

Girls between the ages of 8 and 16 could, with the permission of their parents, join the "Lotta Girls", who were trained for future roles in adult Lotta divisions and entrusted with tasks such as knitting socks and gloves, writing letters to frontline soldiers, and attending the funerals of soldiers killed in action. When these girls turned 17, they could apply to the adult service divisions. However, due to personnel shortages toward the end of the war, "Lotta Girls" aged 14–16 were given more responsibilities and were allowed to participate in some of the more demanding roles usually reserved for adult Lottas. This meant that girls as young as 16 were sent to the front lines to participate in the same activities as their adult peers. Behind the front lines girls were allowed to assist in military hospitals and with preparation of war dead. Comparing these activities to the wartime activities of women who did not volunteer is difficult because the tasks and hardships of women who did not volunteer were so varied. Although we have no data on food consumption during the war, the historic record indicates that the basic needs were met for all citizens, with the exception of short periods of starvation-level caloric intake occurring for some members of the military[31]. Therefore, in terms of caloric intake, the food rations of Lottas were unlikely to have been any better, and perhaps were a bit worse, than those of women on the home front. All women in our analysis were at least 17 years old by the end of the war.

**Data**. Structured interviews of evacuees from Finnish Karelia during World War II were published in a four volume set called "Siirtokarjalaisten tie"[57]. These records were compiled in an effort to record the lives of the Karelian evacuees during World War II. Over 300 individuals were trained to conduct these interviews, which took place between 1968 and 1970. During this time, an effort was made to locate everyone evacuated from Karelia during the war. Each entry in the published books lists the name, sex, date of birth, birthplace, occupation, year of marriage, reproductive records (name, sex, and date of birth of all children) and membership in various organizations, including Lotta Svärd. If they were married, the name,

date of birth, birthplace, and occupation of their spouse are also listed. These books were scanned with optical character recognition software, and additional software was developed (Kaira Core and Natural Language Processing software designed for use with the Finnish language) to digitize and extract these records (see Loehr et al.[58] for more details on data extraction methods and the construction of the database). Overall, there were data on 163,152 individuals, including spouses, but here we focus on a subset of 37,613 women (31,613 of whom had at least one child), all of whom were evacuees, and for whom we had complete and credible records on their year of birth, place of birth, occupation, and years of birth of all their children. Of these individuals, 4261 were listed as members of Lotta Svärd and were between the ages of 12 and 40 in 1939. Finally, we were able to link some of the women in our data by their full names and exact dates of birth to a historical genealogy which uses digitized Finnish church records called "Karjala-tietokanta"[59]. We used these data to find a subset of 2671 women (477 were Lottas) who had at least one full sister and who were between the ages of 12 and 40 in 1940 ($N = 2272$ reproduced of which 359 were Lottas). All R code for analysis, figures, and data selection is publicly available and can be found on Github[60].

**Statistical analysis**. To analyze the reproductive timing and lifetime reproductive success of Lotta Svärd volunteers, we used the rethinking package[35] in R Studio 3.3.3 to run a GLMM regression. Model fitting was performed using Hamiltonian Monte Carlo resampling, which draws samples from the posterior distribution, and was implemented with version 2.12 of Stan[61]. We used Bayesian inference for all statistical analyses, and assessed convergence of the four Markov chains by inspection of the trace plots (see Supplementary Materials: Figs. 6a, b and 7a, b), Gelman–Rubin $R^2$, and an estimate of the effective number of samples. Healthy trace plots generally show good mixing (i.e., the chains crossover each other early and often), stability (they converge on a single parameter estimate ($y$-axis) across iterations ($x$-axis) and tend to remain in that area). In a Bayesian framework, each model conditions data on prior probability distributions and uses Monte-Carlo methods to generate posterior distributions for each of the parameters. The priors are the initial probabilities for the values of each parameter. This type of analysis allows us to compare posterior distributions across occupational categories, age groups and educational backgrounds without relying on specific post hoc tests[36] and averts the need to adjust for multiple comparisons[62]. We are also better able to visualize and interpret differences between parameter estimates relative to a specific value by reporting and displaying the entire posterior distribution for each predictor and showing the highest density intervals (HDI) to reveal the most credible values for each parameter estimate. Here, we assume that a parameter value was credibly different from the baseline if the 95% HDI did not include zero.

To analyze how volunteering for Lotta Svärd impacted reproductive timing and reproductive success, we generated three models. Each was designed to predict three distinct outcomes: *Model 1*: Time to first birth after the war ($N = 31,613$); *Model 2*: Mean birth intervals after the war ($N = 31,613$); and *Model 3*: Total reproduction after the war ($N = 37,613$)(see Supplementary Materials Table 1— right side). To compare the results of these models with models of female reproductive schedules before the war, we used a subset of these same individuals who had given birth to at least one child before the war ($N = 9862$) and used their age at first birth as their time to first birth *Model 1* before the war. For the mean interbirth intervals before the war these sample criteria were even more restricted and were limited to women who had two or more children before the war began ($N = 5603$) in order to be able to accurately calculate a prewar IBI *Model 2*. However, the same women were used to model overall reproduction before the war *Model 3* ($N = 37,613$). In models 1 and 2, we initially included only women who had reproduced, as nonreproductive women cannot, by definition, have mean IBI or time to first birth. Additional models were therefore developed to determine the models sensitivity to excluding nonreproductive women from models 1 and 2 (see Supplementary Materials: Table 3). We also ran each of these models again with all of the same covariates (see Supplementary Materials: Table 2) but this time on a subset of women who we were able to link to a historical genealogy[59]. In these analyses we included all women whose parents were known and who had at least one sister ($N = 2272$ for time to reproduction and mean birth intervals after the war and $N = 2671$ for total reproduction). In this subset, the sisters within a family could either be one Lotta and one non-Lotta, both Lottas or both non-Lottas. For this subset we ran the three models again, but this time included parent id as a random (clustering) intercept to control for within family effects[63]. As described above, the criteria for individuals to be included in the models run to analyze reproductive schedules before the war for the sisters only sample were more restricted for the models used to predict time to first reproduction ($N = 729$) and for mean IBI ($N = 268$), but was the same for the model used to predict overall reproduction ($N = 2671$).

The predictor variables for all analyses were as follows: age when the war ended in 1945 (scaled by subtracting the mean and dividing by the standard deviation of the entire vector using the "scale" function in R 3.5.1[64]), dummy variables encoding whether or not their occupation required an education (binary: 1 = yes, 0 = no), whether or not they were a farmer (binary: 1 = farmer, 0 = not a farmer), whether their first child was born after the war (binary: 1 = yes, 0 = no), whether or not they had given birth within the previous 2 years (binary: 1 = yes, 0 = no), whether or not they had volunteered for Lotta Svärd (binary: 1 = yes, 0 = no), and an interaction between their age in 1945 and whether or not they had volunteered.

Finally, place of birth ($N = 991$) was entered as a random effect into all models. Agriculture and education were entered into the models because previous analyses have shown that these categories explain much of the variance in social status and social integration among this population[32]. "First child born after the war" was used to parse the effects of including women who had already had a child before 1945. For some analyses we replaced this variable with a dummy variable "Married before the war" (see Supplementary Materials Table 7). These two variables could not be entered into the same models because they were highly correlated ($r = 0.70$). However, because "wedding year" was not available for 15,472 women (approximately 41% of our full sample) we only used the dummy variable "Married before the war" in models in which we were primarily concerned with analyzing the effects of being married on reproductive outcomes. The variable "reproduced within the last 2 years" was entered to control for the reduced fertility of women following a birth[65]. The interaction between volunteer status (Lotta) and a woman's age during the war was the predictor of interest.

Statistical analyses for all models were performed in R version 3.3.2 and Bayesian inference used to conduct analyses for Models 1–3 was carried out using the rstan package for R version 2.14.1[66] an interface to Stan which uses a Hamiltonian Monte Carlo sampler[67]. We used the rethinking R package version 1.59[35], which includes convenience functions for building, sampling, and summarizing models with a Bayesian framework[36]. The replicate models using all women, including nonreproductives, used Cox proportional hazards regression models, implemented with the functions coxph and Surv from the survival package [version 2.44-1.1][68]. This allowed us to account for censored data—in this case, right censored at 25 years (the number of years from 1945 to the interviews). Though this may bias estimates upwards for older women, the level of censoring was similar between Lottas and non-Lottas.

A small subset of volunteers in our data identified the specific units to which they were assigned. We created two broad categories based on these identifications that we hoped would capture the level of threat and exposure to mortality that different types of volunteers faced. Canteen workers, nurses and anti-aircraft volunteers were all either stationed nearer to the front lines or spent more time in hospitals and were therefore categorized as "More exposed to combat" while office workers and organizational volunteers spent less time close to combat and hospitals and were therefore categorized as "Less exposed to combat"[69]. We analyzed time to reproduction, IBI and overall reproduction after the war for these two types of volunteers (see "Results").

**Model validity, effects, and specifications**. To assess the validity of these models and their ability to reverse engineer the observed data, we conducted a posterior predictive check (see Supplementary Materials: Fig. 3a, b for the models including the full sample and Fig. 4a, b for the sisters only models). Bayesian models are generative, which means that the posterior distributions produced by these models (see Supplementary Materials: Figs. 1a, b and 2a, b) can be used to make specific predictions on counterfactual data. This also allows us to determine the absolute effect—the practical change in the probability of an outcome occurring that depends on the values of all of the other covariates in the model—that specific parameters of interest have on outcomes. These predictions are generated from the model to construct posterior predictions for a previously unobserved, fictitious, and potentially impossible person. For example, this might be a Lotta Svärd volunteer who is 15 years old when the war breaks out, has a mean education identical to that of our sample, an occupation of average "agriculturalness", and has the mean of the sample values for the dummy variables "reproduced within the last 2 years" and "first child born after the war". These factors are then used by the model to generate predicted posterior distributions. Hamiltonian Monte Carlo Chains, programmed in STAN via the rstan interface, were used to generate these posterior distributions. Broad but weakly regularizing priors that tamp the effects of extreme values were specified for these models as follows: normal distributions of discrete variables were centered on 0, normal distributions of continuously varying covariates were centered on null-hypothesized isometric slopes, and standard deviations were specified as Cauchy distributions with a shape parameter of 1. Models were run with four replicate chains for 6000 MCMC iterations, of which 2000 were warm-up iterations. See Supplementary Fig. 6a, b for trace plots generated by these chains.

**Reporting summary**. Further information on research design is available in the Nature Research Reporting Summary linked to this article.

## Data availability
The data that was used to generate these results and that supports the findings of this study are available on Github: https://github.com/robertlynch66/Lotta-LRS. A reporting summary for this Article is available as a Supplementary Information file. The source data underlying all main article and Supplementary Figures and Tables are provided on Github.

## Code availability
The code used to produce these models, generate all results and produce all of the figures and tables in this manuscript and the supplementary information is available on Github: https://github.com/robertlynch66/Lotta-LRS

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

## Acknowledgements

The Kone Foundation provided funding to R.L. and J.L. R.L. also acknowledges funding from NEFER (decision 321264) and the INVEST research flagship 320162. We would like to thank Jenni Pettay for help for helping to translate much of the source data from Finnish to English and Juuso Kallioniemi and Tuomas Salmi for helping to extract the data from the original texts. The Academy of Finland.

## Author contributions

R.L. wrote the first draft of the paper, conducted the statistical analysis, and made the figures and tables. J.L. and V.L. planned the study and J.L. oversaw the data collection. M.B. and S.C. helped with the statistical analyses. All authors helped to write and edit the paper.

## Competing interests

The authors declare no competing interests.
