## [Peer Review File · Nature Communications]

Reviewers' Comments:

Reviewer #1:

Remarks to the Author:

Review of child volunteers...

This study uses historical data from women evacuated from their homes in Finland during WWII to examine effects of war time experience (military volunteers vs non-volunteers) on subsequent reproduction. This is a very interesting study using a unique data set. The prediction is that young women (adolescents really) who were exposed to presumably greater mortality cues as military volunteers would have earlier reproduction and faster overall pace of reproduction than would non-volunteers who presumably experienced relatively attenuated mortality cues. The results fit into a larger body of research using a variant of life history theory that is, at this point, producing rather robust results for large-scale environmental indicators of difficult and disturbed environments such as warfare. For me the results are fairly convincing; however, there are a few relatively minor issues mostly concerning controls and the nature of the "quasi-natural experiment" that if addressed could strengthen the ms.

My main concern with this study is its characterization as a natural experiment and the relative paucity of control variables. I think the lack of specific controls is tolerable given the historical nature of the data; however, some of the exposition could be tightened up to make a more convincing argument. There are some controls for family occupation in the dataset, but the audience has to take on faith that the volunteers and non-volunteers were otherwise well matched. This gap in the data made me uneasy throughout. Similarly, though this study is framed as a natural experiment it really is not in that the relevant mortality exposure was self-selected by those willing to be military volunteers. We do not know that volunteers and non-volunteers had no other significant differences that might have been relevant. (The authors mention this problem in the discussion but they do not offer a solution or data to counter it.) Was there a difference in whether other family members (especially parents) were volunteers or conscripts? Did they have family members who died in combat prior to or during the evacuation? Etc. And what was the war-time experience of the non-volunteers? I could imagine in some situations that some military volunteers might have had an easier time than other evacuees if they had better rations for example. There was basically one crude socioeconomic control for occupation (a second for farmers). The subset of the analysis using only volunteers and their non-volunteer sisters is more convincing, and in some ways I think this should've been the focus of the paper. It is still not clear how well sisters were matched for age and evacuation status. Were they all evacuated? The very large sample size of the full dataset is attractive, but without controls we cannot take the results as particularly strong support (or a good attempt at falsification) for the prediction. Hence, I think the ms should be reframed around the best (not most plentiful data) the authors have on hand. I also think devoting some space for more historical context, such as the conditions the non-volunteers experienced, would be useful.

I am also concerned by the significant difference between volunteers and non-volunteers who never reproduced. More volunteers never reproduced, and though a hazards model (in the supplement) indicates greater HR of first birth for volunteers vs non-volunteers including childless women, I think some speculation is in order to hypothesize or account for the differences in infertility.

Why were births before 1945 excluded from the analysis? This choice is not at all clear to me.

On a relatively minor note, the ms mentions father-absence repeatedly. Why? There is no data here for father absence. I understand that there has been a resurgence of interest in father-absence among human behavior and evolution scholars; however, much of those more recent analyses indicate that in the developing world (as opposed to developed nations) father absence has little influence on children's long-term reproductive outcomes (e.g. <https://royalsocietypublishing.org/doi/pdf/10.1098/rstb.2018.0124>). It's not clear to me how one

might characterize the development of Finland in 1939, but at any rate it's irrelevant in that there is no data for father absence presented here.

Figures 1, 2 and 3 are little hard to interpret in that observed differences don't seem to match the predicted differences from the model. In fact, the observed differences don't seem significant. Some guidance in the text or figure caption would be welcomed.

Finally, psychosocial acceleration theory (which does not appear to be the focus here) argues that early life adversity during a sensitive period (during the first 7 years of life) for reproductive development influences subsequent early reproduction. I think a few studies have challenged this assumption head on, and those studies find that reproductive strategies are sensitive to conditions throughout life at least up to first birth. It may or may not be worth making that point more forcefully in the discussion.

In sum, I think this is an interesting paper that could be improved with more attention to the best data the authors have available (the sisters subsample), and with more historical context to help assure the audience of the adequacy of the analysis and the nature of the "quasi-natural experiment".

Reviewer #2:

Remarks to the Author:

I believe this to be a worthwhile study that warrants publication after revision, but I must acknowledge at the outset that I'm not familiar with the statistical methods that the authors use. Thus, for example, I have only a vague notion of what the third last sentence of the Methods section ("Broad but weakly regularizing priors that tamp the effects of extreme values were specified for these models as follows: normal distributions of discrete variables were centered on 0, normal distributions of continuously-varying covariates were centered on null-hypothesized isometric slopes, and standard deviations were specified as Cauchy distributions with a shape parameter of 1") means; for all I can tell, it might be gibberish. That said, I think I can nevertheless provide some useful comments, and must hope that other reviewers will be able to speak to the adequacy and appropriateness of these methods.

One reason why I think the study is interesting and of value is that I find its rationale persuasive. There is indeed reason to believe that exposure to cues of mortality risk in youth might shift people's reproductive scheduling in the ways that the authors outline, and it is of interest to not just demonstrate such phenomena, but also to begin to address questions about the critical periods of developmental exposure and the magnitude of the effects. In this regard, the authors' data base seems to be exceptionally rich.

I have one rather large conceptual qualm, however, concerning an issue that the authors have apparently overlooked, and that is the question of whether volunteers might be a selected subset of young women in some relevant way. The paper's very title raises the question - might voluntarism be a confound? - and the reader (or at least this reader) soon discovers that this is indeed a concern, since the crucial comparisons are between women who volunteered for Lotta Svärd and "their peers who did not volunteer". The authors have done an admirable job of addressing the potential confounding effects of various measured variables, but the trouble is that whether one volunteers could conceivably be influenced, net of the effects of all those other measured variables, by a personality variable (boldness? "urgency"?) that also influences age at first reproduction. I don't consider this a fatal problem, but I think that it makes the paper's contribution to the issues of interest more tentative than is acknowledged, and that it is something that the authors must address in discussion.

A distinct issue is that I cannot fathom how any reader could extract meaning from the final "trace

plots" in the Supplementary Materials, and I doubt that that's just because I don't understand the statistical methods behind them. There is no explanation of the significance of the different colours, nor are there captions, at least not in the consolidated manuscript. (The first one's label says "reroduction" in place of "reproduction".) How are these plots useful?

Some additional little comments, mostly stylistic, are these:

Intro 1st sentence: "among" would be better than "between", here and subsequently. Best usage is "between" two things, but "among" three or more.

"Data" is a plural noun. Sometimes, the authors appropriately use a plural verb when "data" is the subject, but sometimes not (e.g. 2nd last line on p 1)

p 2: "variability in the causes of fertility" would be better than just "causes of fertility" in the 2nd sentence of the 2nd full paragraph on this page.

p 2: How is it possible that reference 42 (which is not readily accessible) can be described as "an experimental study"? People were surely not randomly assigned to these different childhood environments?!

p 2: "Regardless of which cues humans pay attention to when adjusting life history strategies" has an excessively cognitive tone ("pay attention"). I think all you mean is "Regardless of which cues humans respond to when adjusting life history strategies". And again, in this paragraph, research is called "experimental" when it surely was not.

Throughout the paper, the present tense is used awkwardly and distractingly where past tense would be appropriate. The war began in 1939 (not "begins") and has ended (not "ends"); various things occurred (not occur) in these women's lives; and so forth.

p 3: 2nd sentence of results section: "Results how..." should be "Results show..."

p 4: "Next, we investigated prediction the prediction that..."

p 5: Figure caption: In the phrase "... a relatively stronger impact...", the word "relatively" is redundant; "stronger" is already relative.

p 6: repetitiveness: The phrase "a natural 'experiment' in which a single population is temporarily split into two groups" is repeated almost verbatim in paragraphs 1 and 2 of the discussion, and was also used in the introduction.

p 7: missing a return before the 2nd Methods section, headed "Data", and then again before each subsequent section heading

and 2 lines later: "subset 3,190 full sisters" requires an "of"

Reviewer #3:

Remarks to the Author:

This is an interesting paper that is well-written and theoretically clear. It shows meaningful, robust results that were derived from good data using strong methods. I do, however, have several comments that the authors should ideally address before publication.

1. My primary concern with this paper is that there should be additional explanation and defense of the idea that there is no selection bias differentiating the girls who joined Lotta Svard from

those who did not. This is important as the paper's claims of having solved analytical problems inherent in other papers rests in part on this distinction. The authors state that there are no differences, but very limited evidence for this is offered. Even if there is some selection bias differentiating the groups the results are still interesting and the methods still justified, but some of the claims to having uniquely strong quasi-experimental methods would be unjustified. To be convincing on the question of selection bias the authors would need to show us side by side statistics on how similar the LS volunteers were to the non-volunteers; this could be put in the supplemental materials. It isn't enough to assert that these are both large populations or that a few controls are used (e.g. education and whether someone was in agriculture). This needs further space and justification up front, though I am glad this concern is acknowledged more fully on page 7.

2. The authors should also specifically address the possibility that the effects they find might be attributed to an alternative explanation. Specifically, it is possible that the differences are due not to greater exposure to mortality but simply to greater exposure to soldiers/men than would have been true if LS volunteers had stayed at home. Greater access to potential mates could account for earlier age at first birth and this could account for greater lifetime RS. Such an effect would be stronger for unmarried women and for younger women, both of whom would likely have had more limited exposure to young men (and thus marriage/mating opportunities) in their home villages than they would as volunteers for LS.

3. Further discussion is needed about what the women in LS were doing during the war, as some jobs (e.g. nurse) may have involved much greater exposure to mortality than would other jobs (e.g. canteen worker), while both may have involved relatively equal exposure to soldiers (see point #2 above). Do the differences between different types of Lotta jobs need to be controlled for and/or examined? Either they should be examined and the differences discussed, or a clear argument should be made for why the different jobs are not examined—you surely have the sample size to do this, so it's hard to see why it is not done given the theoretical implications.

4. The analysis of a subsample of the data with sisters is not fully explained in the text. Why is this a good robustness check? Is the idea that one sister was in LS and one was not? Did the analysis directly compare sisters? Discussion on page 7 suggests this may be the case but the analysis is not fully explained. The rationale needs to be detailed clearly in the text and the implications of the analysis discussed.

Specific comments:

1. In the abstract, the authors state that the research has "implications for efforts to mitigate the adverse effects of childhood adversity on life history outcomes." But the effects described in the paper—a slightly earlier age at first birth and a modestly larger completed fertility—are not adverse in the Finnish circumstances. Given that no particularly negative outcomes are described in the paper, this comment sounds like lip service rather than a meaningful contribution. This wording should be taken out or the text in question restated to more closely match with the direct contributions made by this paper.

2. In the first paragraph of the main paper, I suggest adding the bolded text as follows:
"...including when to start reproduction, how many children to have, and how much to invest in each,..."

3. Page 2, second paragraph: you should consider in this review the potentially differing effects of chronic stress from sudden extreme stress on reproduction in humans. A good discussion of this is given in Nolin & Ziker's 2016 Human Nature paper where the authors document a sudden, rapid fertility reduction in response to the collapse of the Soviet Union (they are not the only ones to document this change—just the most evolutionary). Does exposure to war make sense as a chronic stressor or a more extreme stress? To me it appears that it would be the latter, but your results are more consistent with the classic interpretations of chronic stress.

4. On page 2 the authors state that "Regardless of which cues humans pay attention to when adjusting life history strategies, signals received prior to sexual maturity are expected to have the greatest impact." While I agree that there are a lot of findings that suggest this, there are other findings that suggest that later signals or current environment are more important, or equally important. Thus I think this comment is an overstatement. See for example work by Quinlan comparing the effects of early exposure vs. current environment in Dominica. At the very least this sentence needs to be cited so we know who is making this argument. You might consider Ellis 2004 or other review papers as a place to look for a consensus or cite for this perspective (if they agree with you).

5. In the Results section, how did you determine the cutoffs (e.g. under the age of 20) discussed? In addition, why is the cutoff mentioned in the text "(e.g. under the age of 20)" different from that shown in figure B which shows women "under 19", "19-28" and "over 28". The cutoffs should be justified and also used/discussed systematically.

6. Are durations of exposure to war (e.g. durations of volunteering with LS) controlled for in all of the analyses presented? What other controls are included? This should be clear in the figure legends—especially since the figures come well before the methods section which is where it is discussed in the text.

7. I would be a bit cautious in relying too heavily on Griskevicius et al. 2011; there are at least three attempts to replicate this work (that I know of) which have failed, though not all of them are published (yet). It is certainly reasonable to cite the paper, but you might want to consider additionally citing other similar findings (McAllister et al. 2016 may be a source of useful citations) or mentioning/citing the fact that not all similar tests show the same results.

8. Page 8, middle paragraph: Model 3 should be bolded.

9. The discussion of censored data on the bottom of page 8 is not clear; censored in what way? More specifics on this would allow readers to better follow this discussion.

10. At the bottom of page 8 there is a heading "Model validity, effects, and specification" which is at the end of a paragraph; this looks like it is out of place.

We thank the editor and reviewers for their time and effort with refereeing our manuscript, and for their helpful and constructive comments. Reviewer comments are bolded and in size 12 font, our response is in size 14 font and changes to the text are quoted and italicized.

In particular, each reviewer points out selection issues that remain, despite the nature of your data. While Reviewer 3 suggests providing more data to support your claims that volunteers are similar to the non volunteers, Reviewer 1 suggests focusing on the sister data. Our recommendation is to focus your analyses on the sister data, while incorporating Reviewer 3's request for more sample details. Reviewer 3 also raises an interesting counter explanation for your results: that it is exposure to soldiers that affected reproductive behavior, rather than exposure to mortality. We agree that this is an important issue to resolve, and suggest including the analysis Reviewer 3 suggested. Please address these and the other concerns raised by the reviewers. If you would like to discuss your revision plan, please reach out to me.

In response to the editor's summary comments and those of the reviewers we have made the following three major changes to the manuscript (for more specific changes, see responses to individual reviewer comments):

1) We have shifted the focus of the manuscript to emphasize the sisters dataset as suggested by the editor and reviewer #1 . To accomplish this we have run new models and included these results. We have also removed one of the figures we included in the main manuscript file in the original submission (Figure 2 which showed the mean inter-birth intervals of Lotta volunteers on the Y axis and age on the X axis) and added two more which show the mean time to reproduction (Figure 3) and overall reproduction (Figure 4) for the sisters only analysis. Figure 1 shows the mean time to reproduction and Figure 2 shows overall reproduction for the full samples. The figures and tables in the Supplementary Materials have also been either changed to reflect the new focus on sisters. Figures S2a-b (posterior distribution plots) and S4a-b (posterior predictive checks) for the models using the sisters dataset have been added and we have removed the inter-birth intervals figures (formerly shown in the middle panels (b) of the Supplementary Materials figures showing the posterior distributions and the posterior predictive checks). However the results of these models for inter-birth intervals for the full sample and

sisters only are still shown in tables (see: Supplementary Materials: Tables S1 and S2 - middle panels).

We have also made substantial changes to the text in the main ms and supplementary materials files reflecting this basic change in emphasis (see responses to reviewers below).

2) We have also made substantial changes and run several additional models to address the crucial issue of selection bias brought up by the editor and reviewers #2 and #3. In particular, we have added an analysis of the reproductive schedules of women who eventually volunteered for Lotta Svard BEFORE THE WAR began (see Supplementary materials: Tables S1 and S2 -left side). We believe that this is a major improvement. It goes beyond what reviewer #3 suggested and adds considerable credibility to our results. In addition, as reviewer #3 suggested, we have added Supplementary materials: Table S6 which compares volunteers and non volunteers across a number of potentially relevant traits. Overall, these changes have substantially improved on our ability to make inferences from these data.

3) We have also made substantial changes to address the key issue of whether it is exposure to mortality or stress rather than simply more exposure to men brought up by the editor and Reviewer #3.

We have added Table S5 which shows descriptive statistics for volunteers who were in different units and who had different types of tasks during the war, some of which were more likely to be exposed to mortality than others. As suggested by Reviewer #3, we also have run new models which include a dummy variable (married before or after the war began) and its interaction with volunteering to help parse the effects of being exposed to men and being exposed to mortality (see Supplementary Materials Table S7). This interaction was not included in the initial paper because we only have wedding year for approximately half of the sample. Results of these models indicate that the effect of being a young Lotta on accelerated reproductive schedules is stronger amongst women who were single when the war began, but that the overall effect still holds when we include this

interaction in the models. Overall this suggests that exposure to males may account for some of the reported effects but is unlikely to account for all of it.

Reviewers' comments:

Reviewer #1 (Remarks to the Author):

Review of child volunteers...

This study uses historical data from women evacuated from their homes in Finland during WWII to examine effects of war time experience (military volunteers vs non-volunteers) on subsequent reproduction. This is a very interesting study using a unique data set. The prediction is that young women (adolescents really) who were exposed to presumably greater mortality cues as military volunteers would have earlier reproduction and faster overall pace of reproduction than would non-volunteers who presumably experienced relatively attenuated mortality cues. The results fit into a larger body of research using a variant of life history theory that is, at this point, producing rather robust results for large-scale environmental indicators of difficult and disturbed environments such as warfare. For me the results are fairly convincing; however, there are a few relatively minor issues mostly concerning controls and the nature of the “quasi-natural experiment” that if addressed could strengthen the ms.

My main concern with this study is its characterization as a natural experiment and the relative paucity of control variables. I think the lack of specific controls is tolerable given the historical nature of the data; however, some of the exposition could be tightened up to make a more convincing argument. There are some controls for family occupation in the dataset, but the audience has to take on faith that the volunteers and non-volunteers were otherwise well matched. This gap in the data made me uneasy throughout. Similarly, though this study is framed as a natural experiment it really is not in that the relevant mortality exposure was self-selected by those willing to be military volunteers. We do not know that volunteers and non-volunteers had no other significant differences that might have been relevant. (The authors mention this problem in the discussion but they do not offer a solution or data to counter it.)

We agreed with this concern and have now addressed it in multiple ways. First, to try to address this concern we have added Table S6 to the Supplementary Materials which shows side by side the traits of women who volunteered vs those who did not for the full sample of evacuees and for the subset of sisters. Although this table reveals some of the key advantages (sisters are much closer than non sisters on most of the traits) and *some* disadvantages (e.g. sample size and higher standard

errors) of using the sisters models, it will never be entirely satisfactory. This is because there are still any number of potential differences of which we are unable to take into account without conducting a controlled experiment with random assignment (which is obviously impossible in humans). For example, we do not have genetic data or personality traits (e.g. extraversion) and there are nearly endless varieties of other potential differences between women who volunteer and those who do not that may have an effect on age sensitive reproductive timing. By adopting your suggestion to focus our analysis on the sister's data (see below) we are however 'controlling' for many family effects which is a step in the right direction.

Second, in the previous manuscript, we failed to take full advantage of the natural experiment, which these unique data actually provide. We realized during the revision process that we were able to use the SAME groups of women before the war to effectively act as a pre-treatment group (differential exposure to the war is the treatment) and our analysis of reproductive schedules after the war are the post treatment group. Therefore, we have now added models, which compare the reproductive output and rates for Lottas vs non-Lottas before the war (pre-treatment) as a baseline (see Supplementary materials Tables S1 for full sample and Table S2 for sisters only). The first model (Supplementary Materials: Table S1- right panel), using the full sample of women, shows an age X volunteer interaction (Lotta X age) such that volunteers who were younger had faster reproductive schedules (i.e. shorter time to first birth after the war, post war inter-birth intervals and higher reproduction) after the war ends. By comparison, we have added the results of new models run to predict the reproductive rates and output of the same group of women before the war (Supplementary Materials: Table S1 - left panel). Results show that, although before the war young volunteers did reproduce at a somewhat younger age and had somewhat more children, these effects were less pronounced than they were after the war (see 'Lotta X age' interaction in the top and bottom panels of Supplementary Materials: Table S1) . However, the interaction between age and volunteering was *opposite* for length of inter-birth intervals. Whereas before the war younger volunteers had *longer* inter-birth intervals (Supplementary Materials: Table S1 middle panel, left side), after the war they were *shorter* (Table S1 middle panel, right side).

We repeated this comparison for the sister's only analysis. Here the different reproductive schedules of the same group of young volunteers before and after the

war was *even more notable* (see Supplementary Materials, Tables S2). Before the war, there were no significant differences in the reproductive timing (i.e. age at first reproduction and inter-birth interval: see top and middle panels ‘Lotta X Age’ interaction in Supplementary Materials, Table S2, left side) and younger volunteers had *fewer* children (bottom panel Table S2). Although after the war there was still no detectable effect of an age X volunteer interaction on inter-birth intervals (middle panel, right side Table S2), younger volunteers did wait *less time* to reproduce (top panel, right side Table S2) and have *more* children overall (bottom panel, right side Table S2), consistent with the predictions.

We believe that together these results substantially improve our ability to make inferences from these data. However, it is important to note that, although these comparisons between before and after the war mark a substantial improvement from the previous analyses, they are still imperfect. This is because before the war and after the war sample sizes differ for time to first reproduction (for women who reproduced before the war) and inter-birth intervals. This is for the simple and unavoidable reason that the sample of women who gave birth before the war began had to have given birth before the war in order to have a measure of their before war ‘age at first birth’. The criteria were even more restricted for inter-birth intervals because in this case only women who had at least **two** children before the war were able to be included in the before the war sample. This is because there is no before the war inter-birth interval for women who had less than two children before the war began. Nevertheless there is still substantial overlap between the samples (e.g. all the before war women are included in the post war sample and a dummy variable marking whether they had given birth prior to the war was used in the post war models). But it is just as important to recognize that for overall reproduction bottom panel of Tables S1 and S2 the EXACT same sample of women were used before and after the war so precise comparisons can be made between the models.

Changes to main text:

We added the following sentences to the Methods section in the second paragraph of the ‘*Statistical Analysis*’ subsection:

‘To compare the results of these models with models of female reproductive schedules before the war, we used a subset of these same individuals who had

*given birth to at least one child before the war (N=9,862) and used their age at first birth as their time to first birth **Model 1** before the war. For the mean interbirth intervals before the war these sample criteria were even more restricted and were limited to women who had two or more children before the war began (N=5,603) in order to be able to accurately calculate a pre-war inter-birth interval **Model 2**. However, the same sample of women were used to model overall reproduction before the war **Model 3** (N=37,613).”*

And the following to the end of this same paragraph:

”As described above, the criteria for individuals to be included in the models analyzing pre-war reproductive schedules for the sisters only sample were more restricted for the models used to predict time to first reproduction (N=729) and for mean inter-birth interval (N=268), but were the same for the model used to predict overall reproduction (N=2,671).”

We have added the following subsection ‘Selection bias’ to the Results:

“Selection bias

A persistent problem for quasi-experimental studies like this one, where members of Lotta Svärd are volunteers, is that of selection bias. In this study, this can result in unmeasured differences between women who volunteered and those who did not, which may differentially affect their reproductive timing. In these models, however, we have sought to control for as much of this as possible. First, factors that previous research or theoretical considerations indicated have important impacts on reproductive outcomes, such as education or farming, were included in the models. Second, by analyzing a subset of women who had at least one full sister (see Methods: Statistical Analysis) we were able to take into account many family effects that can affect fertility outcomes such as family size or father absence. However, because there are still any number of potential differences that we are unable to fully take into account, we have included descriptive statistics (means and standard errors), comparing Lottas with Non-Lottas on a variety of traits (e.g. percentage with an education and number of siblings) that previous research has suggested might affect reproduction (see Supplementary materials:

Table S6). These comparisons reveal some of the key advantages and disadvantages of only analyzing sisters. For example, although sisters are much closer than non-sisters on most of the analyzed traits, there is more uncertainty around the estimates.

However, we are also able to take advantage of another opportunity that these data present. Because the war separates the same population of women, we can look upon the different experiences of the volunteers and the non-volunteers during the war as a treatment while regarding their reproductive schedules before and after the war as pre and post-treatment conditions. Therefore we can effectively use models of the reproductive schedules of the same women before the war began as a baseline (i.e. pre-treatment group) with which we are able to compare models of their reproductive schedules after the war (i.e. post-treatment group). Results of models using the full sample of women (see Supplementary Materials: Tables S1) show that, although young volunteers did reproduce at a younger age and had more children also before the war, these effects were considerably less pronounced than they were after the war (see ‘Lotta X age’ interaction in the top and bottom panels of Supplementary Materials: Table S1). The interaction between age and volunteering for length of inter-birth intervals before the war, however, was in the opposite direction, such that younger volunteers had longer inter-birth intervals (Table S1 - middle panel, left side). After the war, they were shorter (Table S1 - middle panel, right side). For the sisters only analysis (Supplementary Materials: Tables S2), these differences were even more notable. Before the war, for instance, there were no detectable differences (95% HDI overlaps with zero) in the age based reproductive timing of Lottas and non-Lottas (see ‘Lotta X Age’ interaction in Supplementary Materials: Table S2 - top and middle panels, left side) and younger volunteers had fewer children overall (bottom panel). In contrast, after the war younger volunteers waited less time to reproduce (top panel, right side) and had more children overall (bottom panel, right side). Even though there was still no detectable effect of an age X volunteer interaction on inter-birth intervals (middle panel, right side), the effect is in the predicted direction. Overall the models analyzing the age based reproductive outcomes of volunteers before the war provide additional support for **Predictions 1-3** that experiences during the war differentially affect the reproductive schedules of young volunteers. “

We have added the following to the third to last paragraph the Discussion section:

“Finally, although modeling the reproductive schedules of women before the war and comparing these results to those of these women after the war marks a substantial improvement over the type of selection bias issues that have plagued previous quasi-experimental studies on natural populations, they are still imperfect. This is because, despite a substantial overlap between the samples (i.e. a subset is used before the war), the sample sizes differ for models assessing reproduction before and after the war for both time to first reproduction and inter-birth intervals (see Methods: 'Statistical Analysis'). Nevertheless, the fact that all of the women are included in the post war sample and a dummy variable marking whether they had given birth prior to the war was used in these models (Supplementary Materials: Tables S1 and S2 -right panels) offers more credibility to the likelihood that exposure to mortality during the war had some effect on these results. Furthermore, it is just as important to recognize that for overall reproduction the exact same sample of women was used both before and after the war (S1 and S2 - bottom panel), making more precise comparisons possible.”

Changes to supplementary materials text:

Table S1 caption now reads: *“Parameter estimates, Highest Density Intervals (HDI's) and Odds ratios for factors affecting time to first reproduction (top panel), mean birth interval after the war (middle panel) and total reproduction (bottom panel) before (left side) and after (right side) the war for full sample. The parameter estimate of primary interest, the Lotta X Age interaction is both more pronounced and in the **hypothesized** direction after the war (i.e. positive or more positive for time to first reproduction and mean inter-birth intervals and negative for total reproduction (right panels). *Parameter estimate 95% HDI does not overlap with zero.”*

Table S2 caption now reads: *“Parameter estimates, Highest Density Intervals (HDI's) and Odds ratios for factors affecting time to first reproduction (top panel), mean birth interval after the war (middle panel) and total reproduction (bottom panel) before (left side) and after (right side) the war for sisters only. The parameter estimate of primary interest, the Lotta X Age interaction is both more pronounced and in the **hypothesized** direction after the war (i.e. positive or more positive for time to first reproduction and mean inter-birth intervals and negative*

*for total reproduction (right panels). *Parameter estimate 95% HDI does not overlap with zero.”*

Was there a difference in whether other family members (especially parents) were volunteers or conscripts?

Yes, and we published a recent paper¹ showing that women with more brothers and husbands who served (but not fathers), were more likely to volunteer. We have added the following to the 2nd sentence of the penultimate paragraph of the Discussion:

“Indeed, previous research on this same population has shown that women who had more brothers and whose husbands served in the military were more likely to volunteer.”

Did they have family members who died in combat prior to or during the evacuation? Etc.

This is a good question but is unfortunately one that we are unable to answer. These data are based on interviews of evacuees who were alive in 1970. Therefore, we have no information on individuals who may have died either during the evacuation or during the war. We also do not have age or cause of death for any of their relatives.

And what was the war-time experience of the non-volunteers? I could imagine in some situations that some military volunteers might have had an easier time than other evacuees if they had better rations for example.

This is a good point, and it is possible that in some cases non-volunteers could have experienced more difficult circumstances than Lotta volunteers. Unfortunately, we do not have any statistics with which to compare the overall nutritional value of food for the military with that of civilians in sufficient detail. However, food rationing was instituted in Finland for the civilian population in 1939, and, of course, the military also had a strict ration regime. There are reports of temporary starvation-level rations for the military due to logistical problems brought on by the transportation and distribution difficulties encountered. For civilians the food rationing system made it possible to attain a sufficient level of

caloric intake and there are no reports of starvation. During the winter of 1941-42 grain reserves would not have been sufficient to feed the population, but starvation was avoided through a pact with Germany, which supplied Finland with sufficient food in exchange for military cooperation. Food availability was probably best for those who could grow their own food. For example, once their own needs were met, farmers were required to hand over food to the rationing system. In practice, however, many farmers kept more than they needed and sold surplus food on the black market or to relatives.

Overall, it would be difficult to conclude that the nutritional status of the military or civilian populations would have differed significantly. It is likely that there was a fair degree of variance in the nutritional levels of individuals in both the military and civilian populations. However, we can conclude that the basic needs were met for all citizens, with the exception of short periods of starvation-level caloric intake occurring for some members of the military. (See response to comment below for changes to manuscript.)

There was basically one crude socioeconomic control for occupation (a second for farmers). The subset of the analysis using only volunteers and their non-volunteer sisters is more convincing, and in some ways I think this should've been the focus of the paper.

We have taken your advice and have reorganized the revision to also include models run on a subset of sisters only. The Introduction, Discussion, Results and Methods sections as well as the figures and tables all reflect this new focus and the following changes have been made to the manuscript:

Changes to main text:

The second to last sentence in the Introduction now reads:

“Finally, we were able to link some of our data to genealogical and interview records to analyze a subset of sisters who came from the same families and those

who specifically identified their wartime units, and therefore the tasks to which they were assigned during the war.”

A third sentence has been added to the first paragraph of the Results section:

“This was true both for models run on the full sample of evacuees [N=37,613 and N=31,613 for all women and only women who reproduced respectively] and for a subset of individuals who we were able to link to a genealogical database and who were from the same families and who had at least one sister [N=2,671 and N=2,272 for all women and only women who reproduced, respectively] (see Methods).”

In the 2nd paragraph of the Results section, we added *“Using the full sample of evacuees [N=31,613]...”* to the beginning of the 2nd sentence and added model predictions for the sisters only model to the end of this paragraph:

“Using a subset of evacuees whose parents were known and who had at least one sister [N=2,272], the model predicts that a volunteer who was 15 years old when the war broke out waited an average of 4.65 (95% PI: 4.25-5.08) years until they had their first child after the war ended. This is two thirds of a year less than the prediction for a 15-year-old who did not volunteer (5.32 years, 95% PI: 4.90-5.75). The opposite pattern is seen for women who were 30 years old when the war began, who were predicted to wait an average of 0.6 years longer -- 4.23 (95% PI: 3.97-4.51) and 3.86 (95% PI: 3.62-4.12) years for Lottas and non Lottas respectively (see Figure 3 and Supplementary Materials: Table S2 -top panel, right side - and Figure S2a).”

In the 3rd paragraph of the Results section, we added *“Using the full sample of evacuees [N=31,613]...”* to the beginning of the 2nd sentence and added model predictions for the sisters only model to the end of this paragraph:

“This prediction, however, received only slight support from the subset of evacuees whose parents were known and who had at least one sister [N=2,272]. Here the model predicts that a volunteer who was 15 years old when the war broke out would have had a mean post-war birth interval of 5.58 (95% PI: 4.39-6.96) years, which is nearly identical to the predicted birth interval of 15-year-old girls who

did not volunteer 5.56 (95% PI: 4.46-6.65) years. Older volunteers (i.e. women who were 30 years old when the war began), however, were predicted to have somewhat longer postwar birth intervals -- 4.07 (95% PI: 3.60-4.55) and 3.85 (95% PI: 3.43-4.30) years for volunteers and non-volunteers respectively (see Supplementary Materials: Table S2 -middle panel, right side) which is consistent with prediction (P2), but does not offer strong support of it.”

In the 4th paragraph of the Results section, we added “Using the full sample of evacuees [N=31,613]...” to the beginning of the second sentence and added model predictions for the sister’s only model to the end of this paragraph:

“We also tested this prediction using a subset of evacuees whose parents were known and who had at least one sister [N=2,671]. Although results were in the predicted direction, they do not offer strong support for the hypothesis. Here the model predicts that a volunteer who was 15 years old when the war broke out would have 1.19 (95% PI: 0.70-1.85) children after the war ends, which is only slightly more than the 1.11 (95% PI: 0.66-1.68) children 15-year-old girls who did not volunteer are predicted to have. A stronger, opposite pattern, however, is seen for older volunteers, whereby volunteers were predicted to have fewer children after the war than non-volunteers -- 0.94 (95% PI: 0.56-1.44) and 1.18 (95% PI: 0.71-1.75) children after the war for 30-year-old Lottas and non Lottas respectively (see Figure 4 and Supplementary Materials: Table S2 -bottom panel, right side - and Figure S2b).”

The Results subsection ‘sister’s analysis’ has been DELETED from the manuscript as a result of these changes.

The Discussion section has been revised as follows:

Overall we have emphasized ‘family effects’ more throughout the Discussion and we added the following sentence - the third sentence of the 2nd paragraph:

“Analyses of the full population-based sample of evacuees, and a subset of women who had at least one sister and controlling for family effects (e.g. shared parents, environments and genetics amongst siblings), yielded similar results.”

We have also added the following to the third to last paragraph of the Discussion:

“It is important to note, for instance, that although within family’s women who volunteered did have both accelerated reproductive schedules and higher overall reproduction as compared to their non Lotta sisters, the effects sizes were not as high as they were with the full sample (see Supplementary Table S2). This does suggest that we should be cautious in over interpreting the strength of our results. At the same time, however, the analysis of sisters does have its disadvantages. The lower effect size found in the sister analysis may be due in large part to a much lower sample size and therefore an overall reduction in statistical power to detect differences. Furthermore limiting our analysis to only include families with at least two daughters can bias the sample by excluding singletons (or those with only brothers) from the analysis. The side by side analysis of the traits of women who volunteered vs those who did not for the full sample of evacuees and for the subset of sisters (Supplementary materials Table S6) also reveals some of the key advantages (e.g. sisters are much closer than non sisters on most of the traits) and disadvantages (e.g. sample size and higher standard errors) of limiting our analysis to only sisters. This is simply due to the limitations of historical datasets like this. For example, we do not have genetic data or personality traits (e.g. extraversion) and a nearly endless variety of other potential differences between women who volunteer and those who do not that may have an effect on age sensitive reproductive timing. Although analyzing sisters and controlling for many potential family effects is a big step in the right direction, there are still any number of possible differences of which we are unable to take into account without conducting a controlled experiment with random assignment. Still it is important to recognize that if this effect were primarily a social one (e.g. Lottas are more gregarious or less inhibited) it is unclear why it would only be manifested in volunteers who were young when the war began.”

The Methods section has been revised as follows:

We have added the following to the end of the 2nd paragraph ‘*Statistical Analysis*’:

“We also ran each of these models again with all of the same covariates (see Supplementary Materials Table S2), but this time on a subset of women who we were able to link to a historical genealogy². Using this subset of women whose parents were known and who had at least one sister (N=2,272 for time to reproduction and mean birth intervals after the war and N=2,671 for total reproduction) we ran the three models again, but also included parent id as a random (clustering) intercept to control for within family effects.”

Figures 3 and 4 have been added to the main text and Figures 2a-b, 4a-b and S7a-b have been added to the supplementary materials to reflect this new focus.

Table S2 also shows the exact parameter estimates and credibility intervals for these model generated posterior distributions.

It is still not clear how well sisters were matched for age and evacuation status. Were they all evacuated?

Yes, everyone in these data were evacuees and the sisters were all evacuees too. We have clarified this in the main ms here:

“Overall there were data on approximately 163,152 individuals, including spouses, but here we focus on a subset of 37,613 women (31,613 of whom had at least one child), all of whom were evacuees and for whom we had complete and credible records on their year of birth, place of birth, occupation, and years of birth of all their children.”

However, the sisters were not matched for age as this would have reduced the sample too much, but age was included as a covariate in all analyses.

The very large sample size of the full dataset is attractive, but without controls we cannot take the results as particularly strong support (or a good attempt at falsification) for the prediction. Hence, I think the ms should be reframed around the best (not most plentiful data) the authors have on hand.

We thank you for raising this point. The revision has been restructured around the sister's analysis, and they are now the primary focus of the paper. However, we still include the full sample of individuals because we believe there are advantages and disadvantages of both.

We have added the following text to the 'selection bias' subsection in Results:

“First, factors that previous research or theoretical considerations indicated have important impacts on reproductive outcomes, such as education or farming, were included in the models. Second, by analyzing a subset of women who had at least one full sister (see Methods: Statistical Analysis) we were able to take into account many family effects that can affect fertility outcomes such as family size or father absence. However, because there are still any number of potential differences that we are still unable to fully take into account so we have included descriptive statistics (means and standard errors), comparing Lottas with Non-Lottas on a variety of traits (e.g. percentage with an education and number of siblings) that previous research has suggested might affect reproduction (see Supplementary materials: Table S6). These comparisons reveal some of the key advantages and disadvantages of only analyzing sisters. For example, although sisters are much closer than non sisters on most of the analyzed traits, there is more uncertainty around the estimates.”

I also think devoting some space for more historical context, such as the conditions the non-volunteers experienced, would be useful.

Okay, we have added the following sentences to the 2nd paragraph of the 'Lotta Svärd organization' subsection in the Methods:

“Comparing these activities to the wartime activities of women who did not volunteer is difficult because the tasks and hardships of women who did not volunteer was so varied. Although we have no data on food consumption during the war, the historic record indicates that the basic needs were met for all citizens, with the exception of short periods of starvation-level caloric intake occurring for some members of the military³. Therefore, in terms of caloric intake, the food

rations of Lottas were unlikely to have been any better, and perhaps were a bit worse, than those of women on the home front.”

I am also concerned by the significant difference between volunteers and non-volunteers who never reproduced. More volunteers never reproduced, and though a hazards model (in the supplement) indicates greater HR of first birth for volunteers vs non-volunteers including childless women, I think some speculation is in order to hypothesize or account for the differences in infertility.

Yes, we have included in the text under sensitivity analysis a reference to a previous paper we published showing that women with young children were less likely to volunteer. The following has been included in the ‘*Sensitivity Analysis*’ subsection under Results:

“This higher probability of Lottas failing to reproduce remains significant ($\square = 0.15 \pm 0.03$, $p < 0.001$) after correcting for age ($\square = 0.10 \pm 0.001$, $p < 0.001$) but this is not particularly surprising because mothers with dependent children at home are less likely to volunteer¹.”

Why were births before 1945 excluded from the analysis? This choice is not at all clear to me.

Pre-war births were excluded but the women who had children before the war were not. Women who gave birth prior to 1945 were given a dummy code value of 1 (birth category), while those whose first birth was after 1945 were given a zero (see Supplementary materials: Tables S1, S2, S3 and S6). However, for the outcome variable (total children produced), we were only interested in the number of children that women produced after the war because we were primarily interested in the effect that differential exposure (volunteering vs. not volunteering) had on reproduction after this exposure. This was also true of the other two dependent variables (i.e. time to first reproduction after the war and mean interbirth intervals after the war) (see Supplementary materials: Tables S1 and S2 - right side).

But, yes, this is a good point and in the revision we have included the results of 6 new analyses which model reproductive schedules before the war (see

Supplementary materials: Tables S1 and S2 - left side). (For more details and all changes made to the manuscript see response to Reviewer #3, first comment).

On a relatively minor note, the ms mentions father-absence repeatedly. Why? There is no data here for father absence. I understand that there has been a resurgence of interest in father-absence among human behavior and evolution scholars; however, much of those more recent analyses indicate that in the developing world (as opposed to developed nations) father absence has little influence on children's long-term reproductive outcomes (e.g. <https://royalsocietypublishing.org/doi/pdf/10.1098/rstb.2018.0124>). It's not clear to me how one might characterize the development of Finland in 1939, but at any rate it's irrelevant in that there is no data for father absence presented here.

Yes, this is true. We do not have any data on father absence, but we do have data on families (e.g. sisters) who both had the same father. We have edited this sentence in the fifth paragraph of the Discussion:

“First, father absence is not expected to have had an important impact on these results, because volunteers from the same families who had the same fathers — sisters — also had accelerated reproductive schedules.”

We have also added the following sentence and the reference you cite to the manuscript in the 2nd paragraph of the Introduction:

“However, a recent meta-analysis suggests these results may be restricted to WEIRD (Western, Educated, Industrialized, Rich and Democratic) populations⁴.”

Figures 1, 2 and 3 are little hard to interpret in that observed differences don't seem to match the predicted differences from the model. In fact, the observed differences don't seem significant. Some guidance in the text or figure caption would be welcomed.

We agree, but these observed differences (and Standard errors) were for each individual year of age while the predicted lines and shading are from the model included all the covariates entered and are also on a continuous scale (i.e. includes ALL ages). A regularly used alternative is to plot the data corrected for the covariates, in which case the model fit and data would match well, but we believe this approach lacks transparency because it involves manipulation of the raw data.

In any case, to reduce the confusion, we have removed the year-by-year means and standard errors of the raw data in panel A as this was largely duplicated in panel B. The main difference is that in panel B it is grouped by age group rather than by individual years as it was in the previous panel A's. We also added the following sentence to the captions of figure 1, 2 and 4 to help clarify this:

“Differences between the model-generated predictions in panel A and the observed data (panel B) primarily result from the impact of covariates entered into the model.”

Finally, psychosocial acceleration theory (which does not appear to be the focus here) argues that early life adversity during a sensitive period (during the first 7 years of life) for reproductive development is influences subsequent early reproduction. I think a few studies have challenged this assumption head on, and those studies find that reproductive strategies are sensitive to conditions throughout life at least up to first birth. It may or may not be worth making that point more forcefully in the discussion.

Yes, we are glad you mentioned this. We think that it is at least worth mentioning that the timing of these triggers (e.g. at which ages does it matter) during development is still debated and have therefore added the following reference and edited the first sentence of the 4th paragraph of the Discussion to read:

“The mechanisms by which harsh environmental conditions are likely to hasten reproduction, or, on a psychological level, increase discounting rates, are currently debated, but exposure to stress⁵⁻⁷ and elevated local mortality rates⁸⁻¹² are both widely seen as strong predictors of earlier reproduction even if the timing of these effects is still debated¹³”

In sum, I think this is an interesting paper that could be improved with more attention to the best data the authors have available (the sisters subsample), and with more historical context to help assure the audience of the adequacy of the analysis and the nature of the “quasi-natural experiment”.

We thank the reviewer for this positive feedback and for the useful comments.

Reviewer #2 (Remarks to the Author):

I believe this to be a worthwhile study that warrants publication after revision, but I must acknowledge at the outset that I'm not familiar with the statistical methods that the authors use. Thus, for example, I have only a vague notion of what the third last sentence of the Methods section ("Broad but weakly regularizing priors that tamp the effects of extreme values were specified for these models as follows: normal distributions of discrete variables were centered on 0, normal distributions of continuously-varying covariates were centered on null-hypothesized isometric slopes, and standard deviations were specified as Cauchy distributions with a shape parameter of 1") means; for all I can tell, it might be gibberish. That said, I think I can nevertheless provide some useful comments, and must hope that other reviewers will be able to speak to the adequacy and appropriateness of these methods.

One reason why I think the study is interesting and of value is that I find its rationale persuasive. There is indeed reason to believe that exposure to cues of mortality risk in youth might shift people's reproductive scheduling in the ways that the authors outline, and it is of interest to not just demonstrate such phenomena, but also to begin to address questions about the critical periods of developmental exposure and the magnitude of the effects. In this regard, the authors' data base seems to be exceptionally rich.

We thank the reviewer for the positive feedback and the useful comments, which we address in detail below.

I have one rather large conceptual qualm, however, concerning an issue that the authors have apparently overlooked, and that is the question of whether volunteers might be a selected subset of young women in some relevant way. The paper's very title raises the question - might voluntarism be a confound? - and the reader (or at least this reader) soon discovers that this is indeed a concern, since the crucial comparisons are between women who volunteered for Lotta Svärd and "their peers who did not volunteer". The authors have done an admirable job of addressing the potential confounding effects of various measured variables, but the trouble is that whether one volunteers could conceivably be influenced, net of the effects of all those other measured variables, by a personality variable (boldness? "surgency"?) that also influences age at first reproduction. I don't consider this a fatal problem, but I think that it makes the paper's contribution to the issues of interest more tentative than is acknowledged, and that it is something that the authors must address in discussion.

Yes, we agree that selection bias is probably the most crucial issue of the paper and the other reviewers and editor have brought this up as well. This is what we have done to help shed more light on this issue:

Changes to main text:

We added the following sentences to the Methods section in the second paragraph of the ‘*Statistical Analysis*’ subsection:

*‘To compare the results of these models with models of female reproductive schedules before the war, we used a subset of these same individuals who had given birth to at least one child before the war (N=9,862) and used their age at first birth as their time to first birth **Model 1** before the war. For the mean interbirth intervals before the war these sample criteria were even more restricted and were limited to women who had two or more children before the war began (N=5,603) in order to be able to accurately calculate a pre-war inter-birth interval **Model 2**. However, the same sample of women were used to model overall reproduction before the war **Model 3** (N=37,613).’*

And the following to the end of this same paragraph:

”As described above, the criteria for individuals to be included in the models run analyzing reproductive schedules before the war for the sisters only sample were more restricted for the models used to predict time to first reproduction (N=729) and for mean inter-birth interval (N=268) but was the same for the model used to predict overall reproduction (N=2,671).”

We have added the following subsection ‘Selection bias’ to the Results:

“Selection bias

A persistent problem for studies of this type is the issue of selection bias. In quasi-experimental studies in which subjects are not randomly assigned, but rather choose which group they will join, there are problems that can result from unmeasured differences between self-selected groups. As joining Lotta Svärd was voluntary, there may be unknown differences between them. In these models we have sought to control for as much of this as possible, and though controlling for many family effects by analyzing a subset of women who had at least one full sister (see Methods: Statistical Analysis) is an important step in this direction, there are still any number of potential differences that we are still unable to fully take into

account. Therefore we have included descriptive statistics (means and standard errors) comparing Lottas with Non-Lottas on a variety of potentially relevant traits for the full sample of evacuees and comparing Lotta sisters with their non-Lotta sisters for the sisters only subset (see Supplementary materials: Table S6). Variables that were seen as important based on previous research or theoretical considerations were included in the main models, and the comparisons reveal some of the key advantages and disadvantages of only analyzing sisters. For example, although sisters are much closer than non sisters on most of the analyzed traits, there is more uncertainty around the estimates. Because this study depends on a historical dataset with certain unavoidable limitations, neither approach is ever likely to fulfill all desired stipulations.

However, we are able to take advantage of another opportunity that these data present. Because the war separates the same population of women, we can look upon the different experiences of the volunteers and the non volunteers during the war as the treatment, while regarding their reproductive schedules before and after the war as pre- and post-treatment conditions. Therefore, we can effectively use models of the reproductive schedules of the same individuals before the war began as a baseline (i.e. pre-treatment group) with which we are able to compare to models of their reproductive schedules after the war (i.e. post-treatment group). Results of models using the full sample of women (see Supplementary Materials: Tables S1) show that, though young volunteers did reproduce at a younger age and had more children before the war, **these effects were considerably less pronounced than they were after the war** (see 'Lotta X age' interaction in the top and bottom panels of Supplementary Materials: Table S1). The interaction between age and volunteering for length of inter-birth intervals before the war, however, was **in the opposite direction**, such that younger volunteers had longer inter-birth intervals (Table S1 - middle panel, left side), whereas after the war they were shorter (Table S1 - middle panel, right side). For the sisters only analysis (Supplementary Materials: Tables S2), these differences were even more notable. Before the war, there were no significant differences in the reproductive timing of Lottas and non-Lottas (see 'Lotta X Age' interaction in Supplementary Materials: Table S2 - top and middle panels, left side), and younger volunteers had fewer children overall (bottom panel). Although after the war there was still no detectable effect of an age X volunteer interaction on inter-birth intervals (middle panel, right side), younger volunteers did wait less time to reproduce (top panel,

right side) and had more children overall (bottom panel, right side). Same as above: I would add a conclusion sentence here”

We have added the following to the third to last paragraph the Discussion section:

“Finally, although modeling the reproductive schedules of women before the war and comparing these results to those of these women after the war marks a substantial improvement over the type of selection bias issues that have plagued previous quasi-experimental studies on natural populations, they are still imperfect. This is because, even though there is substantial overlap between the samples (i.e. a subset is used before the war), the sample sizes differ for models assessing reproduction before and after the war for both time to first reproduction and inter-birth intervals (see Methods: 'Statistical Analysis'). Nevertheless the fact that ALL of the women are included in the post war sample and a dummy variable marking whether they had given birth prior to the war was used in these models (Supplementary Materials: Tables S1 and S2 -right panels) offers more credibility to the likelihood that the exposure to mortality had some effect on these results. Furthermore it is just as important to recognize that for overall reproduction the exact same sample of women were used both before and after the war (S1 and S2 - bottom panel) making more precise comparisons possible.“

Changes to supplementary materials text:

Table S1 caption now reads: *“Parameter estimates, Highest Density Intervals (HDI's) and Odds ratios for factors affecting time to first reproduction (top panel), mean birth interval after the war (middle panel) and total reproduction (bottom panel) before (left side) and after (right side) the war for full sample. The parameter estimate of primary interest, the Lotta X Age interaction is both more pronounced and in the **hypothesized** direction after the war (i.e. positive or more positive for time to first reproduction and mean inter-birth intervals and negative for total reproduction (right panels). *Parameter estimate 95% HDI does not overlap with zero.”*

Table S2 caption now reads: “*Parameter estimates, Highest Density Intervals (HDI's) and Odds ratios for factors affecting time to first reproduction (top panel), mean birth interval after the war (middle panel) and total reproduction (bottom panel) before (left side) and after (right side) the war for sisters only. The parameter estimate of primary interest, the Lotta X Age interaction is both more pronounced and in the **hypothesized** direction after the war (i.e. positive or more positive for time to first reproduction and mean inter-birth intervals and negative for total reproduction (right panels). *Parameter estimate 95% HDI does not overlap with zero.*”

We have also expanded on the previous tables by adding to the Supplementary Materials Tables S1 (left panels), S2 (left panels) which show the results of models of reproductive schedules before the war and Table S6 which shows descriptive statistics comparing non Lottas with Lottas, and Lottas with their non Lotta sisters on a variety of key traits.

Finally, focusing the revised manuscript more on our analysis of the subset of sisters (see responses to reviewers #1 and #3) should also help with this concern.

A distinct issue is that I cannot fathom how any reader could extract meaning from the final "trace plots" in the Supplementary Materials, and I doubt that that's just because I don't understand the statistical methods behind them. There is no explanation of the significance of the different colours, nor are there captions, at least not in the consolidated manuscript. (The first one's label says "reroduction" in place of "reproduction".) How are these plots useful?

Yes we apologize that this was unclear in the original submission. Please note that ‘trace plots’ are entirely a technical thing, namely to show that models have run well and that all the estimates (posterior distributions) are converging on the same values. The titles have all been fixed, two additional ‘trace plots’ (for the sisters only models) have been added to the revised manuscript and the captions and legend have been improved on all 4 of these figures. We have edited all of the captions of these trace plots in the Supplementary materials Figures 6a, 6b, 7a and 7b and clarified their purpose:

For example for Figure 6a now reads:

“Trace plots for chains modeling time to first reproduction after the war using the sisters only sample shows good mixing, convergence and stability across iterations (X-axis). This means that each chain converged on a similar posterior distribution which is generally seen as a characteristic of clean and healthy Markov chains (see Figure S2a and Table S2 (top right) for posterior distributions for all parameters entered into this model.)”

The colors each represent a different chain and this is now shown in the legend on the right side of these figures.

We have also added the following to the Methods section in the ‘*Statistical analysis*’ subsection:

“Model fitting was performed using Hamiltonian Monte Carlo resampling, which draws samples from the posterior distribution, and was implemented with version 2.12 of Stan¹⁴. We used Bayesian inference for all statistical analyses, and assessed convergence of the 4 Markov chains by inspection of the trace plots (see Supplementary Materials: Figures S6a, 6b, 7a and 7b), Gelman–Rubin R^2 , and an estimate of the effective number of samples. Healthy trace plots generally show good mixing (i.e. the chains cross over each other early and often), stability (they converge on a single parameter estimate (Y axis) across iterations (X axis) and tend to remain in that area).”

Some additional little comments, mostly stylistic, are these:

Intro 1st sentence: "among" would be better than "between", here and subsequently. Best usage is "between" two things, but "among" three or more.

Thanks. We have changed this to ‘among’ and all subsequent uses when their were more than 2 categories being compared or referenced.

"Data" is a plural noun. Sometimes, the authors appropriately use a plural verb when "data" is the subject, but sometimes not (e.g. 2nd last line on p 1)

Right. We have corrected this in the example you cite and in all other instances in which ‘data’ were the subject.

p 2: "variability in the causes of fertility" would be better than just "causes of fertility" in the 2nd sentence of the 2nd full paragraph on this page.

This sentence has been edited to read:

“Furthermore, disentangling the other, numerous and variable causes of fertility (e.g. father absence) from the particular variable of interest...”

p 2: How is it possible that reference 42 (which is not readily accessible) can be described as "an experimental study"? People were surely not randomly assigned to these different childhood environments?!

Right. The reviewer is right that this is not an experimental study, as it was based on self-reported preferences. We have deleted the experimental aspect when referring to this study. We have changed the sentence referencing this study to:

*“One study showed that individuals who **claimed to have** been raised in adverse childhood environments (i.e. high mortality rates, low resources) self-reported earlier preferences for ages at first birth and for higher completed family sizes, while individuals who claimed to have been raised in low mortality, high resource environments had the opposite preferences¹⁵.”*

p 2: "Regardless of which cues humans pay attention to when adjusting life history strategies" has an excessively cognitive tone ("pay attention"). I think all you mean is "Regardless of which cues humans respond to when adjusting life history strategies".

Agreed. We have revised this sentence as suggested

And again, in this paragraph, research is called "experimental" when it surely was not.

This was poorly phrased and we neglected to put in the key fact of this study which was that the treatment group was primed with a NY Times article forecasting higher mortality. We have changed this sentence to the following:

“Evidence showing that individuals who report that they were raised in harsh environments expressed preferences for an earlier age at first birth when they were

exposed to cues indicating mortality rates were increasing¹⁶ supports this idea that reproductive decisions are moderated by environments experienced during childhood.”

Throughout the paper, the present tense is used awkwardly and distractingly where past tense would be appropriate. The war began in 1939 (not "begins") and has ended (not "ends"); various things occurred (not occur) in these women's lives; and so forth.

The reviewer is correct. We have adjusted the tense of these sentences accordingly.

p 3: 2nd sentence of results section: "Results how..." should be "Results show..."

We fixed this typo.

p 4: "Next, we investigated prediction the prediction that..."

Fixed.

p 5: Figure caption: In the phrase "... a relatively stronger impact...", the word "relatively" is redundant; "stronger" is already relative.

We have deleted the word relatively.

p 6: repetitiveness: The phrase "a natural ‘experiment’ in which a single population is temporarily split into two groups" is repeated almost verbatim in paragraphs 1 and 2 of the discussion, and was also used in the introduction.

In the Introduction we have changed this to:

“...a quasi-natural experiment in which a population of evacuees is separated by World War II into two groups, each differentially exposed to mortality cues and stress...”

And in the 2nd paragraph of the discussion we have changed this to:

“Here we take advantage of a quasi-natural experiment in which the sudden onset of war divides a population into two groups.”

p 7: missing a return before the 2nd Methods section, headed "Data", and then again before each subsequent section heading.

These formatting errors in latex have been fixed.

and 2 lines later: "subset 3,190 full sisters" requires an "of"

Fixed.

Reviewer #3 (Remarks to the Author):

This is an interesting paper that is well-written and theoretically clear. It shows meaningful, robust results that were derived from good data using strong methods. I do, however, have several comments that the authors should ideally address before publication.

1. My primary concern with this paper is that there should be additional explanation and defense of the idea that there is no selection bias differentiating the girls who joined Lotta Svard from those who did not. This is important as the paper's claims of having solved analytical problems inherent in other papers rests in part on this distinction. The authors state that there are no differences, but very limited evidence for this is offered. Even if there is some selection bias differentiating the groups the results are still interesting and the methods still justified, but some of the claims to having uniquely strong quasi-experimental methods would be unjustified. To be convincing on the question of selection bias the authors would need to show us side by side statistics on how similar the LS volunteers were to the non-volunteers; this could be put in the supplemental materials. It isn't enough to assert that these are both large populations or that a few controls are used (e.g. education and whether someone was in agriculture). This needs further space and justification up front, though I am glad this concern is acknowledged more fully on page 7.

We have added a table to the Supplementary Materials section: Table S6 as you suggested which shows side by side the traits of women who volunteered vs those who did not for the full sample of evacuees and for the subset of sisters. Although this table reveals some of the key advantages (e.g. sisters are much closer than non-sisters on most of the traits) and some disadvantages (e.g. sample size and higher standard errors) of using the sisters models, it will never be entirely

satisfactory. This is simply due to the limitations of the data. For example, we do not have genetic data, or personality traits, such as extraversion, of these women, and there are nearly endless varieties of other potential differences between women who volunteer and those who do not that may have an effect on age sensitive reproductive timing. Although controlling for many family effects by analyzing sisters (see below) is an important step in the right direction, there are still any number of potential differences that we are unable to take into account without conducting a controlled experiment with random assignment. We have now more thoroughly addressed these differences in the discussion section.

However, in the previous manuscript we did not fully understand nor take advantage of the natural experiment, which these data actually provide. We realized during the revision process that would be able to use the reproductive schedules of many, and in some cases all, of these women before the war to act as a pre-treatment group. To put it another way age specific differences in the reproductive schedules of Lottas and non-Lottas before the war is the control, differential exposure to the war is the treatment, and the analysis of age specific differences in the reproductive schedules of Lottas and non-Lottas after the war is the post treatment effect. Therefore, we have now added models which compare the reproductive output and rates for Lottas vs non-Lottas before the war (pre-treatment) as a baseline (see Supplementary materials Tables S1 for full sample and Table S2 for sisters only).

The first model (Supplementary Materials: Table S1- right panel), using the full sample of women, shows an age X volunteer interaction (Lotta X age) such that volunteers who were younger had faster reproductive schedules (i.e. shorter time to first birth after the war, post war inter-birth intervals and higher reproduction) after the war ends. For comparison, we have run new models to predict the reproductive rates and output of the same group of women before the war and added these results to Supplementary Materials: Table S1 - left panel. Results show that, although before the war young volunteers did reproduce at a somewhat younger age and had somewhat more children, these effects were less pronounced than they were after the war (see 'Lotta X age' interaction in the top and bottom panels of Supplementary Materials: Table S1). However, the interaction between age and volunteering was opposite for length of inter-birth intervals. Whereas before the war younger volunteers had *longer* inter-birth intervals (Supplementary Materials:

Table S1 middle panel, left side), after the war they were *shorter* (Table S1 middle panel, right side).

We repeated this comparison for the sister's only analysis. Here the different reproductive schedules of the same group of young volunteers before and after the war was even more notable (see Supplementary Materials, Tables S2). Before the war, there were no significant differences in the reproductive timing (i.e. age at first reproduction and inter-birth interval: see top and middle panels 'Lotta X Age' interaction in Supplementary Materials, Table S2, left side) and younger volunteers had *fewer* children (bottom panel Table S2). Although after the war there was still no detectable effect of an age X volunteer interaction on inter-birth intervals (middle panel, right side Table S2), younger volunteers did wait less time to reproduce (top panel, right side Table S2) and have *more* children overall (bottom panel, right side Table S2).

We believe that these models together substantially improve on our ability to make inferences from these data. However, it is important to note that, although these comparisons between before and after the war mark a substantial improvement from the previous analyses, they are still imperfect. This is because before the war and after the war the sample sizes differ for time to first reproduction and inter-birth intervals. This is for the simple and unavoidable reason that the sample of women who gave birth before the war began had to have given birth before the war in order to have a measure of their before war 'age at first birth'. The criteria were even more restricted for inter-birth intervals because in this case only women who had given birth to at least **two** children before the war were included in the before the war sample. This is because there is no 'before the war mean inter-birth interval' for women who had less than two children before the war began. Nevertheless there is still substantial overlap between the samples such that all the before war women are included in the post war sample. Moreover, a dummy variable marking whether they had given birth prior to the war was used in all of the post war models. But it is just as important to recognize that for overall reproduction (bottom panel of Tables S1 and S2) the exact same sample of women were used before and after the war so precise comparisons can be made between these models.

Changes to main text:

We added the following sentences to the Methods section in the second paragraph

of the ‘*Statistical Analysis*’ subsection:

*‘To compare the results of these models with models of female reproductive schedules before the war, we used a subset of these same individuals who had given birth to at least one child before the war (N=9,862) and used their age at first birth as their time to first birth **Model 1** before the war. For the mean interbirth intervals before the war these sample criteria were even more restricted and were limited to women who had two or more children before the war began (N=5,603) in order to be able to accurately calculate a pre-war inter-birth interval **Model 2**. However, the same sample of women were used to model overall reproduction before the war **Model 3** (N=37,613).’*

And the following to the end of this same paragraph:

”As described above, the criteria for individuals to be included in the models run analyzing reproductive schedules before the war for the sisters only sample were more restricted for the models used to predict time to first reproduction (N=729) and for mean inter-birth interval (N=268) but was the same for the model used to predict overall reproduction (N=2,671).”

We have added the following subsection ‘Selection bias’ to the Results:

Selection bias

A persistent problem for quasi experimental studies like this one, where members of Lotta Svärd are volunteers, is that of selection bias. In this study this can result in unmeasured differences between women who volunteered and those who did not, which may differentially affect their reproductive timing. In these models, however, we have sought to control for as much of this as possible. First, factors that previous research or theoretical considerations indicated have important impacts on reproductive outcomes, such as education or farming, were included in the models. Second, by analyzing a subset of women who had at least one full sister (see Methods: Statistical Analysis) we were able to take into account many family effects that can affect fertility outcomes such as family size or father absence. However, because there are still any number of potential differences that

we are still unable to fully take into account so we have included descriptive statistics (means and standard errors), comparing Lottas with Non-Lottas on a variety of traits (e.g. percentage with an education and number of siblings) that previous research has suggested might affect reproduction (see Supplementary materials: Table S6). These comparisons reveal some of the key advantages and disadvantages of only analyzing sisters. For example, although sisters are much closer than non sisters on most of the analyzed traits, there is more uncertainty around the estimates.

However we are also able to take advantage of another opportunity that these data present. Because the war separates the same population of women, we can look upon the different experiences of the volunteers and the non volunteers during the war as a treatment while regarding their reproductive schedules before and after the war as pre and post-treatment conditions. Therefore we can effectively use models of the reproductive schedules of the same women before the war began as a baseline (i.e. pre-treatment group) with which we are able to compare models of their reproductive schedules after the war (i.e. post-treatment group). Results of models using the full sample of women (see Supplementary Materials: Tables S1) show that, although young volunteers did reproduce at a younger age and had more children before the war, these effects were considerably less pronounced than they were after the war (see 'Lotta X age' interaction in the top and bottom panels of Supplementary Materials: Table S1). The interaction between age and volunteering for length of inter-birth intervals before the war, however, was in the opposite direction, such that younger volunteers had longer inter-birth intervals (Table S1 - middle panel, left side). After the war they were shorter (Table S1 - middle panel, right side). For the sisters only analysis (Supplementary Materials: Tables S2), these differences were even more notable. Before the war, for instance, there were no detectable differences (95% HDI overlaps with zero) in the age based reproductive timing of Lottas and non-Lottas (see 'Lotta X Age' interaction in Supplementary Materials: Table S2 - top and middle panels, left side) and younger volunteers had fewer children overall (bottom panel). In contrast, after the war younger volunteers waited less time to reproduce (top panel, right side) and had more children overall (bottom panel, right side). Even though there was still no detectable effect of an age X volunteer interaction on inter-birth intervals (middle panel, right side), the effect is in the predicted direction. Overall the models analyzing the age based reproductive outcomes of volunteers before the

*war provide additional support for **Predictions 1-3** that experiences during the war differentially affect the reproductive schedules of young volunteers. “*

We have added the following to the third to last paragraph the Discussion section:

“Finally, although modeling the reproductive schedules of women before the war and comparing these results to those of these women after the war marks a substantial improvement over the type of selection bias issues that have plagued previous quasi experimental studies on natural populations, they are still imperfect. This is because, even though there is substantial overlap between the samples (i.e. a subset is used before the war), the sample sizes differ for models assessing reproduction before and after the war for both time to first reproduction and inter-birth intervals (see Methods: 'Statistical Analysis'). Nevertheless, the fact that all of the women are included in the post war sample and a dummy variable marking whether they had given birth prior to the war was used in these models (Supplementary Materials: Tables S1 and S2 -right panels) offers more credibility to the likelihood that the war had some effect on these results. Furthermore, it is just as important to recognize that for overall reproduction the exact same sample of women were used both before and after the war (S1 and S2 - bottom panel) making more precise comparisons possible.”

Changes to supplementary materials text:

Table S1 caption now reads: *“Parameter estimates, Highest Density Intervals (HDI's) and Odds ratios for factors affecting time to first reproduction (top panel), mean birth interval after the war (middle panel) and total reproduction (bottom panel) before (left side) and after (right side) the war for full sample. The parameter estimate of primary interest, the Lotta X Age interaction, is both more pronounced and in the **hypothesized** direction after the war (i.e. positive or more positive for time to first reproduction and mean inter-birth intervals and negative for total reproduction (right panels). *Parameter estimate 95% HDI does not overlap with zero.”*

Table S2 caption now reads: *“Parameter estimates, Highest Density Intervals (HDI's) and Odds ratios for factors affecting time to first reproduction (top panel),*

*mean birth interval after the war (middle panel) and total reproduction (bottom panel) before (left side) and after (right side) the war for sisters only. The parameter estimate of primary interest, the Lotta X Age interaction is both more pronounced and in the **hypothesized** direction after the war (i.e. positive or more positive for time to first reproduction and mean inter-birth intervals and negative for total reproduction (right panels). *Parameter estimate 95% HDI does not overlap with zero.”*

2. The authors should also specifically address the possibility that the effects they find might be attributed to an alternative explanation. Specifically, it is possible that the differences are due not to greater exposure to mortality but simply to greater exposure to soldiers/men than would have been true if LS volunteers had stayed at home. Greater access to potential mates could account for earlier age at first birth and this could account for greater lifetime RS. Such an effect would be stronger for unmarried women and for younger women, both of whom would likely have had more limited exposure to young men (and thus marriage/mating opportunities) in their home villages than they would as volunteers for LS.

Yes, this is a good point and one that we had considered but were unsure of how to analyze effectively but now think we have found a way to gain some insight into this problem.

Because age at birth was rarely missing from these data while wedding year was missing for many individuals (e.g. we lose about 40% of our sample when we include wedding year), we did not include when women were married in any of the models in the previous draft and instead included the dummy variable ‘gave birth before the war’ which may be a very rough approximation of being married. However, to more precisely distinguish the effect of being married vs. being single before the war began, and thereby gain insight into the question of whether it was ‘exposure to men’ or ‘exposure to mortality’ that resulted in accelerated reproductive schedules for Lotta girls, we have decided to include whether or not a woman was married before the war as a dummy variable in additional analyses we have included in the revisions.

These new models do not, however, also include the dummy variable ‘gave birth before the war’, because it was strongly and positively correlated with being married ($r=0.70$). The results of these new models (Supplementary Materials:

Table S7) show that the main effect (i.e. young Lottas have accelerated reproductive schedules) still holds despite entering both married before the war and the interaction between marrying before the war and being a volunteer. But it is important to note that this interaction between marriage and volunteering was a significant predictor of faster reproductive schedules. Overall the new models indicate that the main effect of interest (Lotta X age interaction) is stronger for women who were single before the war began. In other words, greater exposure to males may indeed explain some of these effects, but it is unlikely to explain all of it (as the result holds also for married women). In addition, there are other plausible explanations for why young and married Lottas might not have faster reproductive schedules than young and married women who did not volunteer, including the possibility that they already had children.

Finally, the primary effect of accelerated reproductive schedules is for Lottas who were between the ages of 12 and 20 (see Figures 1-3) while the mean age at marriage in our sample is 26.2 years old (6 to 14 years after the war begins). The long delay between volunteering and marriage for the youngest women in our sample suggests that it is unlikely that meeting and marrying men with whom they were likely to have had more contact can fully explain these results.

Changes to main text:

We added the following sentences to the Methods section under the paragraph ‘Statistical analysis’,

‘First child born after the war’ was used to parse the effects of including women who had already had a child before 1945. For some analyses we replaced this variable with a dummy variable ‘Married before the war’ (see Supplementary Materials Table S7). These two variables could not be entered into the same models because they were highly correlated ($r=0.70$). However, because ‘wedding year’ was not available for 15,472 women (approximately 41% of our full sample) we only used the dummy variable ‘Married before the war’ in models in which we were primarily concerned with analyzing the effects of being married on reproductive outcomes.’

And added the following section ‘*Effect of exposure to male soldiers*’ to the Results section,

“Effect of exposure to male soldiers

To determine whether these effects are driven by greater exposure to mortality or by greater exposure to men (i.e. soldiers), we also analyzed the effect of being married vs. being single before the war began on the reproductive schedules for a subset of Lotta girls whose year of marriage was known (see Methods). The results of models which include the dummy variable 'married before the war' show that the main effect - that young Lottas have accelerated reproductive schedules - still holds even after entering both 'married before the war' and the interaction between being ‘married before the war’ and being a ‘volunteer’. However, it is important to note that the interaction between being married before the war and being a volunteer was also a significant and positive predictor of faster reproductive schedules (see Supplementary Materials: Table S7). Although this does suggest that greater exposure to men could play a role in these results, this effect could also indicate that the effect of war is simply stronger on single women.”

And the following sentences to the 2nd to the last paragraph of the Discussion section,

“It is also unlikely that increased exposure to mortality is the only factor causing young volunteers to have faster reproductive schedules, and young volunteers being more exposed to soldiers is one plausible alternative explanation. Models which included whether or not these women were married when the war began confirmed that the Lotta by age interaction was still a significant predictor of faster reproductive schedules. But they also indicate that the combination of being married and being a Lotta results in faster reproductive schedules after the war (Supplementary Materials: Table S7). This suggests that volunteers who were already married before the war began also had faster reproductive schedules. Because the reproductive outcomes of married women are less likely to be accelerated by more interactions with men, these results are unlikely to be entirely driven by greater exposure to soldiers. In addition, the primary effect of accelerated reproductive schedules is for Lottas who were between the ages of 12 and 20 (see Figures 1-4), while the mean age at marriage in our sample is 26.2

years old (6 to 14 years after the war begins for these women). Therefore it is unlikely that meeting and marrying men with whom they were likely to have had more contact can fully explain these results.”

Changes to supplementary materials text:

And the following caption to the new Supplementary Materials, Table S7:

“Parameter estimates, Highest Density Intervals (HDI's) and Odds ratios for factors affecting time to first reproduction (top panel), mean birth interval after the war (middle panel) and total reproduction (bottom panel) before (left side) and after (right side) the war for individuals for whom wedding year was known. Here the parameter estimates of primary interest are the interactions between Lotta X Married before the war and Lotta X Age. Although the main effect of young Lottas having accelerated reproductive schedules remains (Lotta X Age 95% HDI does not overlap with zero) even after including whether or not women were married when the war began [dummy coded, married before 1940=1, married after 1939=0], it is important to note that the interaction between volunteering and when the women were married (Lotta X Married before the war) is also significant and runs in the opposite direction. This indicates that the effect of accelerated reproduction among young volunteers is more pronounced among women who were single when the war began.”

3. Further discussion is needed about what the women in LS were doing during the war, as some jobs (e.g. nurse) may have involved much greater exposure to mortality than would other jobs (e.g. canteen worker), while both may have involved relatively equal exposure to soldiers (see point #2 above). Do the differences between different types of Lotta jobs need to be controlled for and/or examined? Either they should be examined and the differences discussed, or a clear argument should be made for why the different jobs are not examined—you surely have the sample size to do this, so it’s hard to see why it is not done given the theoretical implications.

We have added the following subsection to the end of the results section in the main text file:

“Lotta type

Another small subset of the women who identified themselves as being members of Lotta Svärd also reported their division (N=2,580). We categorized

these divisions into two groups that we presumed to be more and less exposed to combat (see Methods). Summary statistics show that young Lottas (i.e. under age 25 in 1945) who were differentially exposed to combat have similar reproductive rates and total reproductive output while older Lottas (i.e. over age 25 in 1945) who were more exposed to combat tend to have slower reproductive schedules (see Supplementary materials Table S5). Models were also run using all of the same covariates we used in the main models (see Supplementary Materials Table S1) but limiting the sample to only volunteers whose role in Lotta Svärd was known (N=2,580) and a dummy variable indicating exposure to combat [1=exposed, 0=less exposed]. Although these results revealed a trend in the hypothesized direction (i.e. younger volunteers in areas that were presumed to have more exposure to combat had somewhat faster reproductive schedules), the interaction term between age and exposure to combat was not significant for any of the dependent variables measuring post war reproductive rate.”

And the following text to the discussion section in the main text file:

“Our analysis of different types of volunteers based on their presumed exposure to combat did reveal a trend in the hypothesized direction (i.e. younger women in Lotta units that were more exposed to mortality had faster reproductive schedules). However, our models failed to detect a significant interaction between Lotta exposure and age for any of the three dependent variables assessing reproductive rate after the war. Although our inability to detect significant differences between these Lotta groups can be written off as the result of severe sample size restrictions (less than 10% of lottas reported their units), our assumptions about which lottas were exposed to higher mortality and/or a considerable amount of noise between categories, it does provide additional reasons to be cautious in over-interpreting these results.”

And the following text to the methods section in the main text file:

“A small subset of volunteers in our data identified the specific units to which they were assigned. We created two broad categories based on these identifications that we hoped would capture the level of threat and exposure to mortality that different types of volunteers faced. Canteen workers, nurses and anti-aircraft volunteers

were all either stationed nearer to the front lines or spent more time in hospitals and were therefore categorized as 'More exposed to combat' while office workers and organizational volunteers spent less time close to combat and hospitals and were therefore categorized as 'Less exposed to combat'¹⁷. We analyzed time to reproduction, inter-birth intervals and overall reproduction after the war for these two types of volunteers (see Results)."

And added Table S5 showing these descriptive statistics to the Supplementary materials. We did not, however, show the non significant results of the full models using this sample but would be happy to do so if the reviewer thinks it would be helpful in making these analyses more transparent.

4. The analysis of a subsample of the data with sisters is not fully explained in the text. Why is this a good robustness check? Is the idea that one sister was in LS and one was not? Did the analysis directly compare sisters? Discussion on page 7 suggests this may be the case but the analysis is not fully explained. The rationale needs to be detailed clearly in the text and the implications of the analysis discussed.

Yes, in response to some of the comments above (e.g. regarding selection bias and differences between volunteers and non volunteers) those of Reviewer #1 and the editor we have refocused the paper on the analysis of the subset of sisters.

Here are the key changes we have made to the manuscript:

Changes to main text:

The 2nd to last sentence in the Introduction now reads:

"Finally, we were able to link some of our data to genealogical and interview records to analyze a subset of sisters who came from the same families and those who specifically identified the units and therefore tasks to which they were assigned during the war."

A third sentence has been added to the first paragraph of the Results section:

"This was true both for models run on the full sample of evacuees [N=37,613 and N=31,613 for all women and only women who reproduced respectively] and for a

subset of individuals who we were able to link to a genealogical database and who were from the same families and who had at least one sister [N=2,671 and N=2,272 for all women and only women who reproduced respectively] (see Methods).”

In the 2nd paragraph of the Results section we added “ *Using the full sample of evacuees [N=31,613]...*” to the beginning of the 2nd sentence and added model predictions for the sisters only model to the end of this paragraph:

“Using a subset of evacuees whose parents were known and who had at least one sister [N=2,272], the model predicts that a volunteer who was 15 years old when the war broke out would have waited an average of 4.65 (95% PI: 4.25-5.08) years until they had their first child after the war ended. This is also almost a year less than the prediction for a 15-year-old who did not volunteer (5.32 years, 95% PI: 4.90-5.75). The opposite pattern is seen for women who were 30 years old when the war began who were predicted to wait an average of 0.6 years longer -- 4.23 (95% PI: 3.97-4.51) and 3.86 (95% PI: 3.62-4.12) years for Lottas and non Lottas respectively (see Figure 3 and Supplementary Materials: Table S2 -top panel, right side - and Figure S2a).”

In the 3rd paragraph of the Results section we added “ *Using the full sample of evacuees [N=31,613]...*” to the beginning of the 2nd sentence and added model predictions for the sisters only model to the end of this paragraph:

“This prediction, however, received only slight support when using a subset of evacuees whose parents were known and who had at least one sister [N=2,272]. Here the model predicts that a volunteer who was 15 years old when the war broke out will have a mean post war birth interval of 5.58 (95% PI: 4.39-6.96) years which is nearly identical to the predicted birth interval of 15 year old girls who did not volunteer 5.56 (95% PI: 4.46-6.65) years. Older volunteers (i.e. women who were 30 years old when the war began), however, were predicted to have somewhat longer postwar birth intervals -- 4.07 (95% PI: 3.60-4.55) and 3.85 (95% PI: 3.43-4.30) years for volunteers and non non volunteers respectively (see Supplementary Materials: Table S2 -middle panel, right side) which is consistent with prediction(P2), but does not offer strong support of it.”

In the 4th paragraph of the Results section we added “ *Using the full sample of evacuees [N=31,613]...*” to the beginning of the 2nd sentence and added model predictions for the sisters only model to the end of this paragraph:

“ We also tested this prediction using a subset of evacuees whose parents were known and who had at least one sister [N=2,671]. Although results were in the predicted direction, they do not offer strong support for the hypothesis. Here the model predicts that a volunteer who was 15 years old when the war broke out would have 1.19 (95% PI: 0.70-1.85) children after the war ends, which is only slightly more than the 1.11 (95% PI: 0.66-1.68) children 15-year-old girls who did not volunteer are predicted to have. A stronger, opposite pattern, however, is seen for older volunteers, whereby volunteers were predicted to have fewer children after the war than non volunteers -- 0.94 (95% PI: 0.56-1.44) and 1.18 (95% PI: 0.71-1.75) children after the war for 30-year-old Lottas and non Lottas respectively (see Figure 4 and Supplementary Materials: Table S2 -bottom panel, right side - and Figure S2b).”

The Results subsection ‘*sisters analysis*’ has been DELETED from the manuscript as a result of these changes.

The Discussion section has been revised as follows:

Overall we have emphasized ‘family effects’ more throughout the Discussion and we added the following sentence - the third sentence of the 2nd paragraph:

“Analyses of the full population-based sample of evacuees, and a subset of women who had at least one sister and controlling for family effects (e.g. shared parents, environments and genetics amongst siblings), yielded similar results.”

We have also added the following to the beginning of the third to last paragraph of the Discussion:

“It is important to note, for instance, that although within families women who volunteered did have both accelerated reproductive schedules and higher overall

reproduction as compared to their non Lotta sisters, the effects sizes were not as high as they were with the full sample (see Supplementary Table S2). This does suggest that we should be cautious in overinterpreting the strength of our results. At the same time, however, the analysis of sisters does have its disadvantages. The lower effect size found in the sister analysis may be due in large part to a much lower sample size and therefore an overall reduction in statistical power to detect differences. Furthermore limiting our analysis to only include families with at least two daughters can bias the sample by excluding singletons (or those with only brothers) from the analysis. The side by side analysis of the traits of women who volunteered vs those who did not for the full sample of evacuees and for the subset of sisters (Supplementary materials Table S6) also reveals some of the key advantages (e.g. sisters are much closer than non sisters on most of the traits) and disadvantages (e.g. sample size and higher standard errors) of limiting our analysis to only sisters. This is simply due to the limitations of historical datasets like this. For example, we do not have genetic data or personality traits (e.g. extraversion) and a nearly endless variety of other potential differences between women who volunteer and those who do not that may have an effect on age sensitive reproductive timing. Although analyzing sisters and controlling for many potential family effects is a big step in the right direction, there are still any number of possible differences of which we are unable to take into account without conducting a controlled experiment with random assignment. Still it is important to recognize that if this effect were primarily a social one (e.g. Lottas are more gregarious or less inhibited) it is unclear why it would only be manifested in volunteers who were young when the war began.”

The Methods section has been revised as follows:

We have added the following to the end of the 2nd paragraph ‘*Statistical Analysis*’:

“We also ran each of these models again with all of the same covariates (see Supplementary Materials Table S2) but this time on a subset of women who we were able to link to a historical genealogy \cite{noauthor_undated-eh}. Using this subset of women whose parents were known and who had at least one sister (N=2,272 for time to reproduction and mean birth intervals after the war and N=2,671 for total reproduction) we ran the three models again but also included parent id as a random (clustering) intercept to control for within family effects.”

Figures 3 and 4 have been added to the main text and Figures 2a-b, 4a-b and S7a-b have been added to the supplementary materials to reflect this new focus.

Table S2 also shows the exact parameter estimates and credibility intervals for these model generated posterior distributions.

Specific comments:

1. In the abstract, the authors state that the research has “implications for efforts to mitigate the adverse effects of childhood adversity on life history outcomes.” But the effects described in the paper—a slightly earlier age at first birth and a modestly larger completed fertility—are not adverse in the Finnish circumstances. Given that no particularly negative outcomes are described in the paper, this comment sounds like lip service rather than a meaningful contribution. This wording should be taken out or the text in question restated to more closely match with the direct contributions made by this paper.

Yes we agree. We have revised this to read: *“has implications for understanding the effect of childhood adversity on life-history outcomes”*

2. In the first paragraph of the main paper, I suggest adding the bolded text as follows: “...including when to start reproduction, how many children to have, and how much to invest in each,...”

The first sentence of the first paragraph has been revised in response to this helpful suggestion and now reads:

“Understanding the variation in women’s reproductive scheduling, including when to start reproduction, how many children to have, and how much to invest in each, between individuals, populations, age groups and environments is of considerable interest to researchers across disciplines, including demography, sociology, anthropology and evolutionary biology.”

3. Page 2, second paragraph: you should consider in this review the potentially differing effects of chronic stress from sudden extreme stress on reproduction in humans. A good discussion of this is given in Nolin & Ziker’s 2016 Human Nature paper where the authors document a sudden, rapid fertility reduction in response to the collapse of the Soviet Union (they are not the only ones to document this change—just the most evolutionary). Does

exposure to war make sense as a chronic stressor or a more extreme stress? To me it appears that it would be the latter, but your results are more consistent with the classic interpretations of chronic stress.

This is a good point. We have added the following to the 2nd to last sentence of the 3rd paragraph in the Introduction:

“However, in one indigenous community in Russia birth rates were seen to fall dramatically shortly after the fall of the Soviet Union¹⁸ suggesting that individuals may at times employ a more conservative wait and see strategy while absorbing more information about a novel environment.”

4. On page 2 the authors state that “Regardless of which cues humans pay attention to when adjusting life history strategies, signals received prior to sexual maturity are expected to have the greatest impact.” While I agree that there are a lot of findings that suggest this, there are other findings that suggest that later signals or current environment are more important, or equally important. Thus I think this comment is an overstatement. See for example work by Quinlan comparing the effects of early exposure vs. current environment in Dominica. At the very least this sentence needs to be cited so we know who is making this argument. You might consider Ellis 2004 or other review papers as a place to look for a consensus or cite for this perspective (if they agree with you).

Right, this is also a very good point and our previous sentence was simply an unreferenced assertion. The Quinlan paper referenced focuses on infant mortality rates experienced early in life (i.e. the year of their birth and then again in the year they first gave birth) showed a quadratic association with age at first birth. But we are more interested in the vulnerability to extrinsic mortality perceived by the women themselves. Therefore, we have toned down the statement, changed ‘signals received prior to sexual maturity’ to ‘signals received during childhood’ and provided 2 citations.

“Regardless of which cues humans respond to when adjusting life history strategies, a number of theorists have suggested that signals received during childhood are expected to have the greatest impact.^{19,20}”

5. In the Results section, how did you determine the cutoffs (e.g. under the age of 20) discussed? In addition, why is the cutoff mentioned in the text “(e.g. under the age of 20)”

different from that shown in figure B which shows women “under 19”, “19-28” and “over 28”. The cutoffs should be justified and also used/discussed systematically.

The cutoffs were somewhat arbitrary and have been removed from the manuscript. The ‘B’ panels have been removed from all the figures to save space and because they were redundant (i.e. the same information is conveyed in the main figure).

6. Are durations of exposure to war (e.g. durations of volunteering with LS) controlled for in all of the analyses presented? What other controls are included? This should be clear in the figure legends—especially since the figures come well before the methods section which is where it is discussed in the text.

Unfortunately, we do not know how long women served but in most cases it was likely to have been for the duration of the war. We are not sure if you mean to put this information in the legend or in the caption. But the following is now included in the captions for the 4 main figures:

For example the caption for Figure 1 now reads:

“(see Supplementary Materials: Figure S1a and Table S1 (top panel, right side) for posterior distributions for all covariates and Fig S3a for Posterior predictive check for this model). Differences between the model-generated predictions and the observed data primarily result from the impact of covariates entered into the model.”

7. I would be a bit cautious in relying too heavily on Griskevicius et al. 2011; there are at least three attempts to replicate this work (that I know of) which have failed, though not all of them are published (yet). It is certainly reasonable to cite the paper, but you might want to consider additionally citing other similar findings (McAllister et al. 2016 may be a source of useful citations) or mentioning/citing the fact that not all similar tests show the same results.

Thank you for bringing this to our attention. We have added the following sentence to the 5th paragraph of the Introduction:

“It is important to note, however, that other experimental studies have failed to provide support for this hypothesis and a review article indicates that overall causation has not yet been firmly established²¹. ”

8. Page 8, middle paragraph: Model 3 should be bolded.

Fixed.

9. The discussion of censored data on the bottom of page 8 is not clear; censored in what way? More specifics on this would allow readers to better follow this discussion.

This has been clarified to read:

“For these analyses, we ran a Cox survival model and included all non-reproductive women as right-censored observations at 25+ (the interviews were conducted in 1970 which was 25 years after the end of the war and no further data was collected at this point), rather than excluding them.”

All the other discussions of censored observations also include this type of clarification.

10. At the bottom of page 8 there is a heading “Model validity, effects, and specification” which is at the end of a paragraph; this looks like it is out of place.

Yes, this was a formatting error. We have reviewed the manuscript several more times correcting this one and as many others as we have seen. I think we have caught them all. Thanks.

References

1. Lynch, R., Lummaa, V. & Loehr, J. Self sacrifice and kin psychology in war: threats to family predict decisions to volunteer for a women’s paramilitary organization. *Evol. Hum.*

- Behav.* (2019). doi:10.1016/j.evolhumbehav.2019.06.001
2. Katiha database. *Karjala-tietokantahaku KATIHA - Karjala-tietokantasäätiö* Available at: <http://www.karjalatk.fi/katiha/index.php>. (Accessed: May 2018)
 3. Nevala-Nurmi, S.-L. Girls and Boys in the Finnish Voluntary Defence Movement. *Ennen & nyt* 3–4 (2006).
 4. Sear, R., Sheppard, P. & Coall, D. A. Cross-cultural evidence does not support universal acceleration of puberty in father-absent households. *Philos. Trans. R. Soc. Lond. B Biol. Sci.* **374**, 20180124 (2019).
 5. Coall, D. A. & Chisholm, J. S. Reproductive development and parental investment during pregnancy: moderating influence of mother's early environment. *Am. J. Hum. Biol.* **22**, 143–153 (2010).
 6. Walker, R. *et al.* Growth rates and life histories in twenty-two small-scale societies. *Am. J. Hum. Biol.* **18**, 295–311 (2006).
 7. Yam, K.-Y., Naninck, E. F. G., Schmidt, M. V., Lucassen, P. J. & Korosi, A. Early-life adversity programs emotional functions and the neuroendocrine stress system: the contribution of nutrition, metabolic hormones and epigenetic mechanisms. *Stress* **18**, 328–342 (2015).
 8. Hill, K. Life history theory and evolutionary anthropology. *Evolutionary Anthropology: Issues, News, and Reviews* (1993).
 9. Promislow, D. & Harvey, P. H. Living fast and dying young: A comparative analysis of life history variation among mammals. *J. Zool.* (1990).
 10. Stearns, S. C. The Role of Development in the Evolution of Life Histories. in *Evolution and Development* 237–258 (Springer Berlin Heidelberg, 1982).

11. Stearns, S. C. The evolution of life histories. (1992).
12. Wilson, M. & Daly, M. Life expectancy, economic inequality, homicide, and reproductive timing in Chicago neighbourhoods. *BMJ* **314**, 1271–1274 (1997).
13. Ellis, B. J. & Bjorklund, D. F. Beyond mental health: an evolutionary analysis of development under risky and supportive environmental conditions: an introduction to the special section. *Dev. Psychol.* **48**, 591–597 (2012).
14. Team, S. D. Stan modeling language users guide and reference manual, version 2.14. 0. *Tech. Rep. NAVTRADEVCCEN* (2016).
15. Adair, L. E. *Family planning in context: sensitivity of fertility desires and intentions to ecological cues*. (Kansas State University, 2015).
16. Griskevicius, V., Delton, A. W., Robertson, T. E. & Tybur, J. M. Environmental contingency in life history strategies: the influence of mortality and socioeconomic status on reproductive timing. *J. Pers. Soc. Psychol.* **100**, 241–254 (2011).
17. Paksuniemi, M. & Keskinen, L. The ‘Guardian Group’ of Finland: Socializing Measures in the Little Lotta Organization during the 1930s and 1940s. *Cultural History* **6**, 190–208 (2017).
18. Nolin, D. A. & Ziker, J. P. Reproductive Responses to Economic Uncertainty. *Hum. Nat.* **27**, 351–371 (2016).
19. Thomas Boyce, W. & Ellis, B. J. Biological sensitivity to context: I. An evolutionary–developmental theory of the origins and functions of stress reactivity. *Dev. Psychopathol.* **17**, 271–301 (2005).
20. Belsky, J., Steinberg, L. & Draper, P. Childhood experience, interpersonal development, and reproductive strategy: and evolutionary theory of socialization. *Child Dev.* **62**, 647–670

(1991).

21. McAllister, L. S., Pepper, G. V., Virgo, S. & Coall, D. A. The evolved psychological mechanisms of fertility motivation: hunting for causation in a sea of correlation. *Philos. Trans. R. Soc. Lond. B Biol. Sci.* **371**, 20150151 (2016).

Reviewers' Comments:

Reviewer #1:

Remarks to the Author:

I have reviewed the response to reviewers. The authors have done an excellent job responding to reviewers' concerns, and I think the paper is appropriate for publication in Nature Communications.

Reviewer #2:

Remarks to the Author:

This is a strong revision. The authors have made strenuous and largely successful efforts to respond to the reviewers' concerns, especially the primary issue raised by all readers, namely that of possible selection biases into the Lotta Svärd volunteers group. Concerns about this issue have been mitigated by new emphases on the sisters analysis, on prewar reproduction, and on between-group comparisons of demographics, and the revised manuscript is furthermore improved by its more forthright acknowledgment of the fact that effects of unmeasured between-group differences (e.g. in temperament or personality) can never be completely ruled out. I recommend acceptance of the revised manuscript.

Some further revision is still necessary, however. The reference list is a mess: many citations are incomplete (e.g. #s 1, 5, 11, 13, 36, 37, 53, 59, 60), and others have missing words (e.g. 27, 43) or idiosyncratic formats that are inconsistent with the rest of the list (e.g. 8 and various cases where abbreviations are used here but not there); there are also words in lower case that should be in upper (finland, finnish, chicago). The main text has some anomalies, too: commas where they don't belong and others missing where they do belong; "expectancy's" where "expectancies" is meant (p 2); "life history theory" occasionally rendered with a hyphen between the first two words though usually not. In short, the entire manuscript needs to be proofread with a great deal more care than has yet been shown. I also dislike the continued use of the present tense when past is more appropriate; look, for example, at the title and caption of Figure 1. This is awkward at best, and in places it may even interfere with comprehension.

Reviewer #3:

Remarks to the Author:

The authors have done a thorough job of revising this paper in response to my comments and those of the editor and other reviewers. I have only one remaining comment of any note, which I do think should be clarified before the final manuscript is accepted for publication.

1. I still think that the explanation of the sister analysis is not as clear as it should be. Specifically, it is still not explicit from most of the text in the main paper if these are always one Lotta and one non-Lotta sister, or if both sisters could potentially be Lottas or non-Lottas. This needs to be explicitly stated and explained in the main text the first time that the sister analysis is mentioned and then echoed throughout. Most of the descriptions of the sister say the following: "...subset of evacuees whose parents were known and who had at least one sister..." which does not make the Lotta status of the sisters clear. Yet a few pieces of text hint that this might be comparing Lotta and non-Lotta sisters, e.g. "Figure 3a-b: Younger volunteers of the same age as their sisters who do not volunteer wait less time to give birth after the war ends in 1945." And "Figure 4a-b: Younger volunteers of the same age as their sisters who do not volunteer are predicted by the model to have slightly more children after the war ended in 1945." In addition, the form of the model should be specified: Are both sisters in the model along with a family-level variable as a control for family effects? Or is this a difference-in-difference style model where the difference between the sisters is the outcome variable? Readers should be very clear about these points after reading the manuscript.

We thank the editor and reviewers for their time and effort with refereeing our manuscript, and for their helpful and constructive comments. Reviewer comments are in Arial size 11 font, and our response is bolded in Times New Roman size 12 font. Changes to the text are in quotes and italicized.

Reviewers' comments:

Reviewer #1 (Remarks to the Author):

I have reviewed the response to reviewers. The authors have done an excellent job responding to reviewers' concerns, and I think the paper is appropriate for publication in Nature Communications.

Reviewer #2 (Remarks to the Author):

This is a strong revision. The authors have made strenuous and largely successful efforts to respond to the reviewers' concerns, especially the primary issue raised by all readers, namely that of possible selection biases into the Lotta Svärd volunteers group. Concerns about this issue have been mitigated by new emphases on the sisters analysis, on prewar reproduction, and on between-group comparisons of demographics, and the revised manuscript is furthermore improved by its more forthright acknowledgment of the fact that effects of unmeasured between-group differences (e.g. in temperament or personality) can never be completely ruled out. I recommend acceptance of the revised manuscript.

Some further revision is still necessary, however. The reference list is a mess: many citations are incomplete (e.g. #s 1, 5, 11, 13, 36, 37, 53, 59, 60), and others have missing words (e.g. 27, 43) or idiosyncratic formats that are inconsistent with the rest of the list (e.g. 8 and various cases where abbreviations are used here but not there);

We have reviewed the references and made all the changes the reviewer mentions and some others.

there are also words in lower case that should be in upper (finland, finnish, chicago). The main text has some anomalies, too: commas where they don't belong and others missing where they do belong; "expectancy's" where "expectancies" is meant (p 2); "life history theory" occasionally rendered with a hyphen between the first two words though usually not. In short, the entire manuscript needs to be proofread with a great deal more care than has yet been shown. I also dislike the continued use of the present tense when past is more appropriate; look, for example, at the title and caption of Figure 1. This is awkward at best, and in places it may even interfere with comprehension.

All co-authors have proofread the manuscript again to improve sentence structure and clarity. We have also corrected any grammatical errors, typos, and spelling mistakes we

found, plus we have standardized our use of hyphenation, and changed the use of present tense to past when necessary. All of the figure titles have been removed, because this information was available in the figure captions.

Reviewer #3 (Remarks to the Author):

The authors have done a thorough job of revising this paper in response to my comments and those of the editor and other reviewers. I have only one remaining comment of any note, which I do think should be clarified before the final manuscript is accepted for publication.

1. I still think that the explanation of the sister analysis is not as clear as it should be.

Specifically, it is still not explicit from most of the text in the main paper if these are always one Lotta and one non-Lotta sister, or if both sisters could potentially be Lottas or non-Lottas. This needs to be explicitly stated and explained in the main text the first time that the sister analysis is mentioned and then echoed throughout. Most of the descriptions of the sister say the following: "...subset of evacuees whose parents were known and who had at least one sister..." which does not make the Lotta status of the sisters clear. Yet a few pieces of text hint that this might be comparing Lotta and non-Lotta sisters, e.g. "Figure 3a-b: Younger volunteers of the same age as their sisters who do not volunteer wait less time to give birth after the war ends in 1945." And "Figure 4a-b: Younger volunteers of the same age as their sisters who do not volunteer are predicted by the model to have slightly more children after the war ended in 1945." In addition, the form of the model should be specified: Are both sisters in the model along with a family-level variable as a control for family effects? Or is this a difference-in-difference style model where the difference between the sisters is the outcome variable? Readers should be very clear about these points after reading the manuscript.

Yes, this is a very good point, and we agree that this was not clear. We included all women whose parents were known (i.e. those we were able to link to a genealogical database) and who had at least one sister, and then added parent id as a random effect to the models. So, no, these were not matched pairs of families with one Lotta sister and one non-Lotta sister: both sisters could potentially be Lottas or non-Lottas. We have now clarified this in the methods section (see below).

Changes to main text:

We removed ‘*as compared to their non-Lotta sisters*’ from the 4th sentence of the 6th paragraph of the discussion.

We edited the 2nd to last sentence of the subsection ‘Data’ in the Methods section:

“*We used these data to find a subset of 2,671 women (477 were Lottas) who had at least one full sister and who were between the ages of 12 and 40 in 1940 (N=2,272 reproduced of which 359 were Lottas).*”

We deleted “*and who were from the same families*” from the third sentence of the first paragraph in the Results section.

The first sentence of the figure captions for figure 3a-b now reads:

“Younger sisters who were the same age as sisters who did not volunteer (i.e. women in families with at least two daughters) waited less time to give birth after the war ended in 1945 in models that included parent id as a random effect (see Methods).”

The first sentence of the figure captions for figure 4a-b now reads:

“Younger sisters who were the same age as sisters who did not volunteer (i.e. women in families with at least two daughters) had slightly more children after the war ended in 1945 in models that included parent id as a random effect (see Methods).”

We modified the second to last sentence of the second paragraph of the ‘Statistical Analysis subsection under Methods:

“In these analyses we included all women whose parents were known and who had at least one sister (N=2,272 for time to reproduction and mean birth intervals after the war and N=2,671 for total reproduction). In this subset, the sisters within a family could either be one Lotta and one non-Lotta, both Lottas, or both non-Lottas. For this subset we ran the three models again, but this time included parent id as a random (clustering) intercept to control for within family effects.¹”

References

- 1.** Zuur, A., Ieno, E. N., Walker, N., Saveliev, A. A. & Smith, G. M. *Mixed effects models and extensions in ecology with R.* (Springer Science & Business Media, 2009).

Reviewers' Comments:

Reviewer #3:

Remarks to the Author:

Thanks to the authors for addressing my comments on the last revision; this has really helped clarify the form of the sisters analysis.

My only remaining comment is that I don't find the following figure caption very intuitive:

“Younger sisters who were the same age as sisters who did not volunteer (i.e. women in families with at least two daughters) waited less time to give birth after the war ended in 1945 in models that included parent id as a random effect (see Methods).”

****REVIEWERS' COMMENTS:**

Reviewer #3 (Remarks to the Author):

Thanks to the authors for addressing my comments on the last revision; this has really helped clarify the form of the sisters analysis.

My only remaining comment is that I don't find the following figure caption very intuitive:

“Younger sisters who were the same age as sisters who did not volunteer (i.e. women in families with at least two daughters) waited less time to give birth after the war ended in 1945 in models that included parent id as a random effect (see Methods).”

Yes, this was a terribly confusing sentence. Thank you for pointing it out. We have changed this caption to clarify:

“In the sisters only analysis (see Methods: Statistical Analysis) the effect of volunteering on time to reproduction after the war was also age specific such that younger volunteers waited less (and older volunteers more) time to give birth than their sisters who did not volunteer.”

And because similar wording was used in the caption for figure 4a-b we also changed this caption to:

“In the sisters only analysis (see Methods: Statistical Analysis) the effect of volunteering on total reproduction after the war was also age specific such that younger volunteers had slightly more (and older volunteers slightly fewer) children than their sisters who did not volunteer.”